# Exploring the inorganic composition of the Asian Tropopause Aerosol Layer using medium-duration balloon flights

*Hazel Vernier[1], Neeraj Rastogi[2], Hongyu Liu[3,4], Amit Kumar Pandit[3], Kris Bedka [4], Anil Patel [2], Madineni Venkat Ratnam[5], Buduru Suneel Kumar[6], Bo Zhang[3], Harish Gadhavi[2], Frank Wienhold[7], Gwenael Berthet[1], Jean-Paul Vernier [3,4]*

1. Laboratoire de Physique et Chimie de l'Environnement et de l'Espace (LPC2E), France

2. Physical Research Laboratory, Ahmedabad, India

3. National Institute of Aerospace, Hampton, VA, USA

4. NASA Langley Research Center, Hampton, VA, USA

5. National Atmospheric Research Laboratory, Gadanki, India

6. TIFR Balloon Facility, Hyderabad, India

7. ETH, Zürich, Switzerland

Correspondence to: Hazel. Vernier ([hazel.vernier@cnrs-orleans.fr](hazel.vernier@cnrs-orleans.fr))

**Abstract.** Satellite observations have revealed an enhanced aerosol layer near the tropopause over Asia during the summer monsoon, called the Asian Tropopause Aerosol Layer (ATAL). In this work, aerosol particles in the ATAL were collected with a balloon-borne impactor near the tropopause region over India, using extended duration balloon flights, in summer 2017 and winter 2018. Their chemical composition was further investigated by quantitative analysis using offline ion chromatography. Nitrate ($NO_3^-$) and nitrite ($NO_2^-$) were found to be the dominant ions in the collected aerosols with values ranging between 87-343 ng/m$^3$ STP (Standard Temperature and Pressure) during the summer campaign. In contrast, sulfate ($SO_4^{2-}$) levels were found above the detection limit (>10 ng/m$^3$ STP) only in winter. In addition, we determined the origin of the

air masses sampled during the flights through analysis of back trajectories along with convective proxy from cloud top temperature fields derived from a geostationary satellite. The results obtained therein were put into a context of large-scale transport and aerosol distribution with GEOS-Chem chemical transport model simulations. The first flight of summer 2017 which sampled air mass within the Asian monsoon anticyclone (AMA), influenced by convection over Western China, was associated with particle size diameters from 0.05 to 0.15 µm. In contrast, the second flight sampled air masses at the edge of the AMA associated with a larger particle size radius ($> 2\mu m$) with a higher nitrite concentration. The sampled air masses in winter 2018 were likely affected by smoke from the Pacific Northwest fire event in Canada, which occurred 7 months before our campaign, associated with concentration enhancements of $SO_4^{2-}$ and $Ca^{2+}$. Overall, our results suggest that nitrogen-containing particles represent a large fraction of cloud-free and in-cloud aerosols populating the ATAL, partially in agreement with the results from aircraft measurements during the StratoClim campaign. The exact nature of those particles is still unknown but their coincidences with subvisible cirrus clouds and their sizes suggest Nitric Acid Trihydrate (NAT) as a possible candidate since already been observed in the tropical upper troposphere and lower stratosphere. Furthermore, GEOS-Chem model simulations indicate that lightning $NO_x$ emissions could significantly impact the production of nitrate aerosols sampled during the summer of 2017.

## 1. Introduction

1.1 Asian Summer Monsoon and the transport of pollution

Over the past two decades, rapid economic growth in Asia has led to serious environmental threats to water and air quality. Every winter, pollutants can be observed through satellites in the form of a grayish veil of particulate matter referred to as the Asian Brown Cloud (Ramanathan and Crutzen, 2003). In summer, the Southwest Asian Monsoon (SAM) discharges polluted air over very long distances. According to trajectory calculations, about 20% of air masses in the tropical lower stratosphere have been in contact with air in the boundary layer in Asia (Orbe et al., 2015). Polluted air masses transported from the boundary layer to higher altitudes are confined within the Asian Monsoon Anticyclone (AMA) (Ploeger et al., 2017). In the AMA, pollution is accumulated and is further dispersed over a large area of the Northern Hemisphere reaching longitudes from 10°-

140°E, and latitudes from 10°- 40°N (Park et al., 2007; Randel et al., 2010; Ungermann et al., 2016). The air exported from the AMA influences the composition of the entire lowermost stratosphere of the Northern Hemisphere (Ploeger et al., 2017; Santee et al., 2017; Yu et al., 2017). Deep convective clouds represent conduits for air pollution to reach the Upper Troposphere and Lower Stratosphere (UTLS) region. Aerosols in the UTLS have longer residence times than those in the lower troposphere, influencing the chemistry of the atmosphere and the Earth's climate (Rasch et al., 2008). In addition, they also affect the concentration of chemical species through changes in photolysis rates and heterogenous reactions (Pitari et al., 2014). It has been further reported that aerosols in the UTLS can impact climate by altering the properties of cirrus clouds via homogeneous or heterogeneous ice nucleation (Li et al., 2005; Liu et al., 2009; Yin et al., 2012; Fadnavis et al., 2013; Wagner et al., 2020).

A layer of aerosol enhancements observed by the Cloud-Aerosol Lidar and Pathfinder Satellite Observations (CALIPSO) and the Stratospheric Aerosol and Gas Experiment (SAGE) II (Vernier et al., 2011; Thomason and Vernier, 2013), also known as the Asian Tropopause Aerosol Layer (ATAL), coincide with the presence of enhanced trace gas pollutants (carbon monoxide (CO), hydrogen cyanide (HCN), etc.) in the UTLS region. Balloon-borne measurements (Vernier et al., 2015, 2018) confirmed the presence of the ATAL at altitudes of 14-18 km, connected to the AMA. The positive trend in UTLS aerosols inferred from satellites observations since the late 90s may reflect the increasing influence of anthropogenic emissions on stratospheric aerosol levels. Indeed, global chemical transport model simulations suggest that sulfate, nitrate, and organic aerosols produced from gas-phase precursors populate the UTLS region over Asia in various relative fractions during the summer monsoon (Brabec et al., 2012; Gu et al., 2016; Fairlie et al., 2020).

1.2 What is the significance of ATAL's composition?

The ATAL constitutes one of the most important sources of UTLS aerosols in the absence of volcanic eruptions (Vernier et al., 2011). It has the potential to affect the Earth's radiative balance (Vernier et al., 2015), stratospheric ozone chemistry, and the properties of cirrus clouds. For example, an increase in solid particle concentration relative to the liquid background aerosol levels could trigger heterogeneous freezing and the formation of cirrus clouds at a lower relative humidity with respect to ice (Cziczo et al., 2015; Wang et al., 2020). Model simulations suggest that the ATAL represents 20% of the total column surface area density in the stratosphere of the Northern

Hemisphere (Yu et al., 2018) with potential halogen heterogeneous chemistry on aerosols that can affect ozone trends (Solomon et al., 2016). The types of aerosols populating the ATAL could affect those chemical processes. Finally, the presence of absorbing aerosols (e.g., soot) in the UTLS could shift the level of zero net radiative heating upward and enhance troposphere-to-stratosphere transport (Yu et al., 2015).

### 1.3 What is known about ATAL's composition?

The composition of the ATAL is a very active research topic. Energy-dispersive X-ray analysis (EDX) of aerosols sampled near 10-12 km onboard commercial aircraft as part of the Civil Aircraft for the Regular Investigation of the atmosphere Based on an Instrument Container (CARIBIC) program, at the bottom part of the ATAL, suggests a ratio between carbon and sulfur in the range 2-10 (Vernier et al., 2015). Aircraft Limb InfraRed measurements carried out during the StratoClim campaign in Nepal and India show the presence of ammonium nitrate in aerosol particles, validating satellite observations from the Cryogenic Infrared Spectrometers and Telescopes for the Atmosphere (CRISTA), and Michelson Interferometer for Passive Atmospheric Sounding MIPAS (Höpfner et al., 2019). A combination of community models and aerosol climate chemistry model indicates that along with surface-emitted and secondary organic aerosols, the ATAL could be comprised of a significant amount of mineral dust either as a major component (Fadnavis et al., 2013; Lau et al., 2018; Ma et al., 2019; Bossolasco et al., 2020) or minor component (Yu et al., 2015; Gu et al., 2016; Yu et al., 2017; Fairlie et al., 2020).

The aerosol particles in the ATAL are looked upon as an insignia of the presence of pollution in the monsoon circulation from large $SO_2$ and $NO_x$ emissions in South and SE Asia. Human-induced biomass burning (Van der A et al., 2008), fossil fuel combustion (Ghude et al., 2009; Bouman et al., 2002), wildfires (Goode et al., 2000; Andrae and Merlet, 2001), and lightning (Martin et al., 2007) are the significant anthropogenic, and natural sources of $NO_x$. Soil biogenic emission of $NO_x$ represents a large fraction of total $NO_x$ (Jalié et al., 2004). Reactive nitrogen is emitted from the tropical soils by microbial processes as NO (Yienger and Levy, 1995; Conrad et al., 1996). Investigations of the composition of the aerosol particles in the ATAL are exiguous, although preliminary data from balloon-borne measurements indicate the presence of nitrate aerosol

particles (Vernier et al., 2018). Recent in situ aerosol mass spectrometric measurements also reveal the presence of nitrate, ammonium, and sulfate within the ATAL (Höpfner et al., 2019).

Here, we investigate the inorganic composition of the ATAL over India during the summer monsoon and winter using a balloon-borne aerosol impactor system with offline Ion Chromatography (IC) analysis. Section 2 describes the concept of the balloon experiment and the impactor system. The IC analysis of the samples collected during two balloon flights in 2017 and on the ground, as well as that of winter 2018 is described in Section 3. Section 4 compares those results obtained from balloon-borne measurements and satellite observations. Section 5 describes the influence of the Canadian wildfire event on the BATAL winter flight. The origin of the air masses sampled during those flights is assessed in section 6 through back-trajectory analysis combined with convective proxies. Section 7 addresses the formation of nitrite and its measurements. The GEOS-Chem model simulations are presented in Section 8 to put the measurements in the context of regional aerosol transport and distribution, followed by a summary and conclusions in Section 9.

**2. Balloon flights, instrumentation, and chemical analysis approach**

*2.1 Rationale for the experiment*

Contingent on measurements during the 2015 Balloon-borne measurement campaigns of the Asian Tropopause Aerosol Layer (BATAL) campaign, a concentration of about 20 particle/$cm^3$ was found near the tropopause for aerosol radius greater than 75 nm (Vernier et al., 2018). It translates into a mass concentration of 40 ng/$m^3$STP (hereafter STP is assumed when mass concentrations are given) assuming that the aerosols were liquid sulfate droplets. During that time, the lower detection limit for the IC instrument at NASA Langley Research Center was around 20 ng/$m^3$. In order to reach the detection limit of sulfate aerosols, one would need to sample at least 0.5 $m^3$ assuming the sulfate concentration above. Based on those results and weight limitations, we decided to use an impactor with a flow rate of 7 lpm which would need to float in the UTLS region for several hours to sample sufficient air volume (2 hours of sampling = 0.84 $m^3$).

*2.2 Balloon experiment*

We used zero-pressure plastic balloons to achieve a float near the tropopause and sample enough aerosols to reach the detection limit of the IC. The Tata Institute of Fundamental Research Balloon

Facility (TIFR-BF) in Hyderabad, India provided the infrastructure to conduct the experiment. 300 to 500 $m^3$ polyethylene balloons manufactured by TIFR were used for the Zero-Pressure flights (ZF) to carry a communication/control package developed by TIFR, a science module including a meteorological radiosonde, a Compact Optical Backscatter and Aerosol Detector (COBALD) (Vernier et al., 2015; Yu et al., 2017), an aerosol impactor, and a ballast module at the end of the flight train. A schematic diagram shown in Fig.1 (top panel) describes a typical balloon flight. During the ascent, atmospheric pressure decreases allowing gas inside the balloon to occupy a large space (stage 2). The equilibrium point is reached when the hydrogen escapes from the side escape tubes attached at the bottom of the balloon, until the inside pressure equals the outside pressure (stage 3) leading to the pressure differential to 0 (zero-pressure balloon). The float altitude depends upon the volume of the balloon, the density of gas, as well as the total weight of the system following simple Archimedes principle.  Extreme cold temperatures near the tropopause affect the float due to radiative cooling, leading to a reduction of the buoyancy force, which entrain the descent of the system (stage 4). To counterbalance this effect, ballast shots are released from a container to reduce the total weight (stage 5) leading to the ascent of the balloon.

*2.3 Balloon-borne Aerosol Impactor*

We developed the Balloon-borne Aerosol Impactor (BAI) for the ZF flights. This aerosol sampler is comprised of a 4-stage impactor, a vacuum pump, a volumetric flow controller, and a Raspberry-PI based controller connected to a meteorological sonde. The mechanical part of the impactor was designed by *California Measurements, Inc*. and is based upon the principle of inertia, where the flow and the instrument dimension determine the size cutoff at different stages.  The size cutoff in radius for the impactor's 4 stages (S1, S2, S3, and S4) is 2, 0.5, 0.15, and 0.05 μm at 7 lpm. The pump is controlled electronically based on the pressure measurements from the meteorological sonde. Our objective is to sample aerosols within the ATAL region, to achieve this the pump was switched on below 150 hPa (~14 km) and switched off above 70 hPa (~18 km). However, due to a reduction of the pump efficiency at those levels, the flow rates lay between 5 and 6 lpm leading to a small shift in size cut-off up to 18% (e.g. 2.36 μm instead of 2 μm for a flow of 5 lpm for S1).

In 2017, we conducted a series of balloon flights using the BAI together with a COBALD sonde for aerosol backscatter measurements of cloud and aerosol layers encountered by the BAI. The time-height evolution of the 3 ZFs is shown in Fig.1 (bottom), with flight ZF1 being a test flight

to understand and/maintain the float altitude using ballast. The maximum flight duration was obtained through ZF3 with a float time of nearly 2h 50min above 150 hPa and below 70 hPa. The oscillation of the balloon trajectories is due to the cooling of the gas inside the balloon and the subsequent release of ballast to regain higher altitudes. The BAI was preserved in a foam box containing dry ice, during transportation to TIFR where the filters were immediately unloaded and stored in 47 mm Petri dishes which were frozen at -24 °C until further analysis at Physical Research Laboratory, Ahmedabad, India. Fig. 2 represents the time evolution of altitude, temperature, and relative humidity inside the box containing the impactor where the different phases of the experiment are mentioned.

### 2.4 Analysis of major ions in aerosol samples

Aerosol samples were extracted in deionized water (Milli-Q, specific resistance $\geq 18.2$ MΩ. cm) in sterile polypropylene vials for 30 minutes (3 intervals of 10 minutes each) using ultrasonication. The extract was further analyzed for water-soluble inorganic species (WSIS, such as $Na^+$, $K^+$, $Mg^{2+}$, $Ca^{2+}$, $NH_4^+$, $Cl^-$, $NO_2^-$, $NO_3^-$, and $SO_4^{2-}$) using an ion chromatograph (IC model-Dionex ICS-5000 DC-5). For calibration, 1000 mg/L stock solution of each cation (using Merck high purity analytical grade $NaNO_2$, $(NH_4)_2SO_4$, $KNO_3$, $CaCl_2.2H_2O$, and Mg metal) were mprepared. In addition, mixed standards were prepared by diluting stock solutions in polypropylene vials, thus satisfying the primary requirement of instrument calibration for cations. Similarly, anion multi-element standard-II (1000 mg/L in $H_2O$, HC 409399, Merck) was diluted subsequently as instrument calibration for anions. Post extraction, the extract of each sample was then separated and eluted in the cation column (DIONEX IonPac$^{TM}$ SC16, $5 \times 250$ mm), and anion column (DIONEX Ion Pac$^{TM}$ AS23, $4 \times 250$ mm) via the interaction with the mobile phases, i.e., 30 mM methyl sulphonic acid (MSA) for cation and a mixture of 4.5 mM carbonate + 0.8 mM bi-carbonate solutions for anions. The quantification of each ion was then performed using the conductivity detector. Several blanks were also analyzed in the same way as the sample, and blank corrected from ionic concentrations are reported. As the concentrations of different species were too low in UTLS aerosol samples, only those values which were at least two times higher than their respective blanks are reported. More than 50% of samples were repeated for reproducibility and found to vary between 2 to 20% for all the analyzed ions. To validate the analysis, Dionex six cation-I standard (product number 040187) and Dionex seven anion standard-II (Part #57590)

1   were diluted and checked in the respective cation and anion calibration curves which were found

2   within ±10% relative standard deviation (RSD).

*3. Results of IC analysis*

Figure 3 shows the concentration of ions from the ground (GND), and two ZF2 (15[th] Aug.), and

ZF3 (21[st] Aug.) flight samples collected during the summer 2017 campaign, in comparison with

the only flight results of the winter 2018 campaign (ZF Winter). In GND samples, $Na^+$ and $Ca^{2+}$

cations are seen on S1 and S2 with corresponding anions ($NO_3^-$, $SO_4^{2-}$, and $NO_2^-$) co-existing at

the same stage. High $NH_4^+$ is observed only on S3 with a concentration of 212 $ng/m^3$ STP. $K^+$

was also seen on S3 with a concentration of 26 $ng/m^3$ STP (fine mode) that could have originated

from biomass burning. City pollution from Hyderabad is ~~likely~~ the source of those aerosols

observed on the GND filters. Flight ZF2 and ZF3 show significant amounts of $NO_3^-$ and $NO_2^-$

(87-343 $ng/m^3$ STP) with traceable amounts of proxies for mineral dust ($Ca^{2+}$). Biomass burning

($K^+$) was observed in the results of flight ZF2 only. The presence of non-sea-salt-$Ca^{2+}$ in aerosols

is often used as a proxy for mineral dust (Schüpbach et al., 2013), and non-sea-salt-$K^+$ in

aerosols is a proxy for biomass burning (Li et al., JGR, 2003). Although their concentrations

were too low (close to the detection limit), their presence indicates a possibility of traces from

mineral dust and biomass burning.

Other species were below 5 (for cations) to 10 (for anions) $ng/m^3$ STP, the detection limit of the

IC instrument for our analytical setup. Charge balance was not achieved due to a higher negative

charge mainly from $NO_3^-$ and $NO_2^-$ than the positive charge mainly from $NH_4^+$, $Ca^{2+}$, and $K^+$ (Fig.

3), implying the existence of $NO_3^-$ and $NO_2^-$ in other forms rather than salt. For instance, nitric

acid trihydrate (NAT, $HNO_3 \cdot 3H_2O$) could be another aerosol cluster in which $NO_3^-$ may be present

in the tropical UTLS (Voigt et al., 2000). We did not find a significant amount of ammonium in

our ZF flight samples during the summer. Overall, the concentration of nitrate (80-100 $ng/m^3$STP)

found on both flights seems to be lower than the levels observed during StratoClim (Höpfner et

al., 2019). In the only flight during the winter of 2018, $Na^+$ and $K^+$ were almost inexistent. In

comparison, the proxy of mineral dust ($Ca^{2+}$) was present on all four stages with traceable amounts

and could be associated with $SO_4^{2-}$ which was also found on all 4 stages (Fig. 3 Bottom).

Balloon-borne and aircraft sampling techniques have been used since the early 70's to study the composition of aerosols in the UTLS region (Lazarus et al., 1970). While sulfate tends to be stable enough to be collected and further analyzed without major chemical transformation, other nitrate-containing particles can be more unstable. $NO_3^-$ salts apart from $NH_4NO_3$ are not significantly volatile after sampling (Leonard Newman, 1993). The dissociation of $NH_4NO_3$ into gas-phase $HNO_3$ and $NH_3$ increases sharply with increasing temperature and relative humidity (Seinfeld et al., 1982; Lightstone et al., 2000), leading to a significant loss of particulate nitrate (PN). The slight retention of $HNO_3$ (gas) on the PTFE filter could represent a significant source of particulate nitrate on filters at low concentrations and was used in the past to estimate stratospheric $HNO_3$ (Lazarus et al., 1970). Additional information available during ZF2 will be discussed to assess the presence of ice clouds.

## *4. COBALD and CALIOP point to the presence of ice clouds during ZF2*

We will now focus the discussion on ZF2 which included a COBALD backscatter sonde and was launched to be collocated in space and time (within 20 km and 1h) with satellite observations from the CALIOP lidar onboard the CALIPSO satellite. Fig. 4 (Top) shows Scattering Ratio (SR) and Color Index (CI) profiles from COBALD (470 nm and 940 nm) together with CALIOP SR and Volume Depolarization profiles at 532 nm. Both balloon and satellite observations show a layer between 13.5 and 16 km with high depolarization (CALIOP) and high color ratio (COBALD), likely made of aspherical particles. The derived particulate depolarization ratio from CALIOP level 2V4.1 within the layer being 0.47+/-0.06 (Fig. S3) with an associated optical depth of 0.03+/0.02 indicates the presence of a subvisible cirrus cloud. Flight ZF2 floated near 14.5-17 km for more than 2h (Fig.1, bottom). The time series (Fig.4, bottom) indicate that the measurements took place within two different air masses: The first within an ice cloud as discussed above, followed by a cloud-free region. The pump connected to the impactor was switched on below 150 hPa and run ~16 min within the cloud and around 1h30min in a cloud-free region.

4.1. In-cloud nitrate particles

The sampling within an ice cloud (Fig. 4) during ZF2 could therefore indicates the presence of in-cloud $NO_3^-$. HNO3, (an oxidation product of NOx) and NH3 (released from agricultural sources) are said to be absorbed into cloud droplets which then aid in the conversion of HNO3 to

1   aerosol $NO_3^-$ (Hayden et al., 2007). HNO3 being readily soluble, tends to completely dissolve in

2   cloud water (Steinfeld and Pandis, 1998). ZF2 sampled 90 ng/m$^3$ STP $NO_3^-$ of particle size (2 µm

3   - 0.5 µm) on stage 2 and 11 ng/m$^3$STP of $NO_3^-$on stage 3 corresponding to particle size (0.5 µm -

0.15 µm). If nitrate enters clouds from the gas phase as an acid, it has to be buffered by $NH_3$ in

order to remain in the aerosols after water evaporation. The buffering process results in nitrate

naturalization, leading to aerosol nitrate formation through cloud cycling (Hayden et al., 2007).

The GEOS-Chem chemical transport model (CTM) showed the presence of inorganic nitrate

aerosol to be dominating the ATAL (Gu et al., 2016). The authors concluded that gas-aerosol

conversion of HNO3 was the driving factor for this dominance by the processes discussed above.

4.2. NAT particles

Another candidate for the presence of nitrate on the filters could be NAT particles. They have been

reported in tropical ice clouds by Voigt et al. 2008 with sizes (d<6 µm) consistent with their

sampling on stages 1 (> 2 µm) and 2 (0.5-2µm) of our impactor. In addition, NAT nucleation

seems to be more efficient in subvisible ice clouds at higher ambient temperature than the

temperature associated with NAT formation at -78°C (Voigt et al., 2008). The sampling within the

ice cloud at temperatures between -65°C and -75°C would allow the presence of NAT. However,

In the process of sampling, transport, and extraction, there is a strong possibility of NAT particle

losses, if they were collected. In addition, if  $NO_3^-$ was present in another form (refractory nitrate)

then they remain relatively stable during the said processes. Observed cations were close to or

below the detection limit compared to the significant concentrations of $NO_2^-$ and $NO_3^-$. This

observation along with the higher abundance of $NO_2^-$ allowed us to suggest the presence of NAT

particles. However, NAT reported concentrations should be considered as the lower limit,

presuming some losses (unquantifiable) during the sampling, transport, and extraction processes.

4.3. In-cloud calcium and its implication

The IC results of flight ZF 2 showed the presence of particles of $Ca^{2+}$ (9 ng/m$^3$STP) on stage 2

(0.5-2µm). Erosion of calcareous soils followed by strong convective vertical transport during

summer results in cloud water calcium (Issac et al., 1990). Cloud water experiments have shown

the formation of Ca(NO3)$_2$ in presence of NH3. Hill et al. (2007) and  Leaitch et al. (1986) found

a positive correlation between $Ca^{2+}$ and $NO_3^-$. In addition to $Ca^{2+}$, ZF2 also showed the presence

of $NO_3^-$ (90 ng/m$^3$STP) consequently at the same stage (large particles < 2µm), further implying the possibility of the formation of Ca(NO3)$_2$ in presence of the acid, HNO3. Lastly, a high concentration of nitrite (193 ng/m$^3$STP) was also found on stage 2 of the impactor. The presence of nitrite in clouds is further discussed in section 7.

## *5. Influence of Canadian wildfire plumes during the winter flight*

In the only flight during the winter of 2018, $Na^+$ and $K^+$ were almost inexistent. In comparison, the proxy of mineral dust ($Ca^{2+}$) was present on all four stages with 30 ng/m$^3$of particle size >2 µm on stage 1, followed by 46 ng/m$^3$ corresponding to particle size between 0.5-2 µm on stage 2. Stage 3 showed 11ng/m$^3$ of particle size between 0.15-0.5 µm and finally stage 4 showed 29 ng/m$^3$ of Ca $^{2+}$ corresponding to particle size between 0.05-0.15 µm. Interestingly, $SO_4^{2-}$ was also found on all 4 stages (Fig. 3 Bottom) with 14 ng/m$^3$ on stage 1, followed by 21 ng/m$^3$ on stage 2. Stage 3 and 4 showed concentrations of 15 ng/m$^3$ and 12 ng/m$^3$ corresponding to particle size between 0.15-0.5 µm and 0.05-0.15 µm respectively. Satellite analysis of aerosol extinction at 1020 nm from the Stratospheric Aerosol and Gaz Experiment III (SAGE III) was conducted to understand the origin of those particles. We found high values of aerosol extinction in the Northern Hemisphere from August 2017 to February 2018 consistent with the presence of smoke from the 2017 Canadian fire (Fig S2).

The 2017 Canadian wildfire event led to the formation of multiple PyroCb episodes resulting in a vast aerosol cloud. Within a few weeks, a portion of this initial plume was transported by the Polar jet streams across the Atlantic Ocean in the northern hemisphere (Peterson et al., 2018) resulting in a strong perturbation of stratospheric aerosol loads (Stocker et al., 2021). The quantity of smoke injected was enormous to the point at which it was observed for more than 8 months (Yu et al., 2019). The presence of the resultant aerosol layer was pointed out by high ultraviolet aerosol index values and confirmed with CALIOP lidar observations in the UTLS (Torres et al., 2020). The aerosol mass increase and subsequent adiabatic aerosol self-lofting as a result of absorption of solar radiation were also observed by the Earth Polychromatic Imaging Camera (EPIC) sensor onboard the Deep Space Climate Observatory (DSCOVR) satellite. Kloss et al (2019) used SAGE III aerosol extinction values to show that the fire plume was transported within the AMA circulation in August 2017. Our analysis suggests that the smoke plume was still present at 18 km

above Hyderabad between January and February 2018 indicating that aerosols sampled during the winter flight were influenced by this smoke plume.

## *6. Convective influence*

Deep convection, emanating from Southeast Asia, and maritime convection over surrounding seas serve as a conduit for the transport of Boundary Layer (BL) pollutants (CO, HCN, $CH_4$) to the UTLS (Bergmann et al., 2013; Park et al., 2007; Randel et al., 2010; Park et al., 2006). Wind-driven physical processes lead to the accumulation of pollutants due to the limited exchanges of air between the interior and exterior of the Asian Monsoon Anticyclone (Fairlie et al., 2014; Ploeger et al., 2015; Fairlie et al., 2020).

 To study the impact of convection on our measurements, we calculate back-trajectories from ZF2 and ZF3 using the Langley Trajectory Model (LaTM; Fairlie et al., 2014) driven by winds from the NASA Global Modelling and Assimilation Office (GMAO) Goddard Earth Observing System, Version 5, Forward Processing (GEOS-5 FP; Lucchesi, 2018). We locate the intersection with anvils and deep convective clouds observed through Cloud Top Brightness Temperature from the HIMAWARI-8 satellite (Vernier et al., 2018). Figure 5 shows the position of those 5-day back-trajectories (colored lines) and deep convection influence (black dots). Air sampled during ZF2 on 15[th] August 2017 traveled along two branches influenced by convection over southern/eastern China and western China, respectively. Air masses sampled by ZF3 originated from convection over Laos, Myanmar, the Bay of Bengal, and possibly local convection over the Indian Eastern Shore close to the measurement location.

## *7  Nitrite measurements*

The role of clouds on nitrite formation is further discussed in this section. Only a few nitrite measurements have been reported to date mainly because of its low concentrations and also because nitrite ions are easily oxidized (Lammel and Cape, 1996). The first quantitative information on nitrite in cloud water was detected at Mt. Tsukuba, Japan. Values of 400-1050 µg/L with pH levels of 5.7-6.5 were reported. In contrast, acidic cloud water samples (pH of 3.4-4.3) collected at significantly higher altitudes showed low nitrite values (15-104 µg/L) (Okita, 1968). Nitrite was also measured in fog water samples in a polluted region in Germany (Lammel

and Metzig, 1998). Moreover, Bachmann et al. (1989 directly measured nitrite in rain and fog water samples using ion chromatography. Values of 1.8 and 16 µmol/L, (86 and 736µg/L), respectively, were found. Photolysis of particulate nitrate, hydrolysis of NO2, and uptake of $HNO_2$ by particles are the sources of particulate nitrite in the atmosphere (Chen et al. 2019).

HNO2 is an important precursor for nitrite formation but there are challenges in making reliable $HNO_2$ measurements at desired concentrations leading to the lack of information about $HNO_2$ in the troposphere. This is mainly due to the rapid reduction of $HNO_2$ during analysis. Secondly, $HNO_2$ being sticky in nature may be lost to the walls of sampling tubes or absorbed on filters. Because nitrite is present in very low concentrations, and the fact that it is easily oxidized, there is limited information on nitrite measurements in the atmosphere, where nitrite and nitrous acid are short-lived intermediates of reactive oxidized nitrogen (Lammel and Cape 1996).

Intensive agricultural activities have led to maximum ammonia ($NH_3$) loading over the Indo Gangetic plains globally (Wang, T. et al. 2019) as revealed by satellite observations (Van Damme et al. 2018; Warner et al. 2016), and ground-based measurements (Carmichael et al. 2003). Nitrite and nitrate are formed by the oxidation of $NH_3$ through the process of nitrification $[NH_3 + O_2 \rightarrow NO_2^- + 3H^- + 2e^-]$. In addition, the existence of NH3 in the presence of nitrate leads to the formation of ammonium nitrate which could neutralize aerosol particles and favor the persistence of nitrite as revealed by a few existing measurements in the polluted region (Lammel and Metzig, 1998). The StratoClim campaign also revealed the presence of ammonium nitrate in the UTLS which would confirm that neutralization of nitrate is effective at high altitudes and may explain the persistence of nitrite found with our balloon measurements.

## *8. Comparison with GEOS-Chem simulations*

We conducted GEOS-Chem model simulations to put our observations in the context of large-scale transport and distribution of atmospheric composition. GEOS-Chem is a state-of-the-art global 3-D chemical transport model that includes fully coupled ozone-$NO_x$-VOC-aerosol chemistry for both troposphere and stratosphere (Bey et al., 2001; Park et al., 2004; Eastham et al., 2014). We use here the model version 11-01 (http://wiki.seas.harvard.edu/geos-chem/index.php/GEOS-Chem_v11-01). A previous version of the model was used to study the

origins of aerosols in the ATAL by Gu et al. (2016) and Fairlie et al. (2020). The model simulates black carbon (Park et al., 2003), primary and secondary organic aerosols (SOA; Pye et al., 2010), sulfate-nitrate-ammonium aerosol thermodynamics coupled to ozone-NOx-hydrocarbon-aerosol chemistry (Park et al., 2004), mineral dust (Fairlie et al., 2007; Ridley et al., 2014), and sea salt (Jaegle et al., 2011), treated as an external mixture. SOA uses the volatility-based scheme (VBS) of Pye et al. (2010). Sulfate-ammonium thermodynamics is computed using the ISORROPIA-II thermodynamic equilibrium model of Fountoukis and Nenes (2007). Aerosol wet deposition includes rainout and washout due to large-scale precipitation as well as scavenging in convective updrafts (Liu et al., 2001). Scavenging of aerosols by snow and mixed precipitation is described by Wang et al. (2011, 2014). Dry deposition of dust and sea-salt aerosols uses the size-dependent scheme of Zhang et al. (2011). Dry deposition for other aerosols follows the resistance-in-series scheme of Wesely (1989). Anthropogenic emissions use the EDGAR database (Olivier & Berdowski, 2001), with regional options, including the MIX inventory over East Asia (Li et al., 2014) and the EPA/NEI 2011 inventory over North America (Travis et al., 2016). Biofuel emissions are from Yevich and Logan (2003). Carbonaceous aerosol emissions are provided by Bond et al. (2007). Biogenic emissions are calculated by the MEGAN model (Guenther et al., 2012). Biomass burning emissions use the Quick-Fire Emissions Dataset (QFED; Darmenov & da Silva, 2015). Lightning $NO_x$ emissions ($LNO_x$) are as described by Murray et al. (2012) and match the Lightning Imaging Sensor and the Optical Transient Detector (LIS/OTD) climatological observations of lightning flashes. Volcanic $SO_2$ emissions are provided by the AeroCom project (data available from www.geos-chem.org). The model simulations are driven by the Modern-Era Retrospective analysis for Research and Applications (MERRA-2) reanalysis from the NASA Global Modeling and Assimilation Office (Gelaro et al., 2017). For computational efficiency, MERRA-2 fields have been mapped from the native grid to $2.5^{o}$(Lon) by $2^{o}$(Lat) horizontal resolution for input to GEOS-Chem. Further, we used the simulations with and without lightning $NO_x$ emissions to understand the contribution of lightning to the formation of nitrate aerosol.

In situ chemical analysis are compared with GEOS-Chem simulations. Fig. 6 shows the maps of CO, nitrate, sulfate, ammonium, black carbon (BC), and dust aerosol concentrations averaged over 100-150 hPa at 22 UTC for Aug. 15[th] and 21[st], respectively, during ZF2 and ZF3 flights (white circle on the map). On Aug. 15[th] CO, BC, nitrate, ammonium, and dust aerosol concentrations are

enhanced over West China, Nepal, and northeastern India with the center of the anticyclone positioned over West China. On the contrary on Aug. 21$^{st}$ during ZF3, the position of the anticyclone was shifted to the east and the flight apparently sampled air at the edge of the anticyclone. A 19% decrease in CO concentration is seen, while a 50 % increase in BC mass on 21$^{st}$ August compared to 15$^{th}$ August. Additionally, Ammonium concentration was decreased by 50% and dust by 60% on Aug.21$^{st}$ compared to the fisrst flight on Aug. 15$^{th}$. However, SO4 concentration is seen to be stable at ~80ng/m$^3$ for both 15$^{th}$ and 21$^{st}$ above Hyderabad. The simulated NO3 concentrations near the location of ZF2 and ZF3 are spatially inhomogeneous with variations between 30 and 2700 ng/m3 across South India. Figure 7 shows the time series of model 3-hourly CO, sulfate, and nitrate concentrations averaged over 100-150hPa within the model grid-point where Hyderabad is located during August 2017. CO concentration shows a decrease of 14%, while an increase of 21% in SO4 concentration is seen in the ZF3 flight held on 21$^{st}$ August. The measured nitrate concentration during ZF2 and ZF3 around ~100 ng/m$^3$ is within the ranges of values simulated within 24h of the observations. The results of the GEOS-Chem model simulation indicate that lightning NO$_x$ could significantly (up to ~75% on August 10th) contribute to the formation of nitrates during certain time periods. The lifetime of NO$_x$ is approximately 3h in the region of the outflow of thunderstorms due to the production of methyl proxy nitrate and alkyl, and multifunctional nitrates and its lifetime is believed to increase downwind from the outflow (Nault et al., 2017). Also shown in Fig.7 are nitrate concentrations attributed to lightning as determined by the difference between simulations with and without lightning NOx emissions. ZF2 and ZF3 occurred during a period where the levels of nitrate were ≈50 ng/m$^3$ STP on 15$^{th}$ August and ≈30 ng/m3 STP on 21$^{st}$ August. There was the minimal influence of lightning NO$_x$ emissions. Nevertheless, CO levels are slightly higher during ZF2 (80 ppbv) than ZF3 (60 ppbv), indicating that measurements made during ZF2 may have been more influenced by pollution. The latter is also reflected by the higher BC levels during ZF2 in the model.

We extracted CO, nitrate, and sulfate concentration from the GEOS-Chem simulation along the calculated trajectories initialized from ZF2 and ZF3 measurement locations in Fig. 8. The lines are colored according to the balloon GPS altitudes which are used to initialize the trajectory model. Fig.5 uses GEOS-FP winds (meteorology) to convey that GEOS-Chem could simulate convective activities reaching levels between 14-15 km. This is confirmed by cloud-top heights (black circle) derived from HIMAWARI-8 crossed by trajectories originating from the

troposphere for both ZF2 and ZF3. ZF2 was influenced by convective activities over Western

China while ZF3 sampled air masses originated from convection in SE Asia (Myanmar, Laos).

CO levels with initial altitudes near 14-15 km (green color) for ZF2 are shown to decrease from

120 ppbv to 80 ppbv along the back-trajectories confirming the influence of Chinese pollution

and its progressive dilution. At the same initial altitudes, the CO levels along ZF3 back-

trajectories are significantly lower near 50-80 ppbv possibly indicating minimal impacts of

polluted sources. The levels of $NO_3^-$ show significant variability along the trajectories for both

cases but are more pronounced in ZF3 with levels above 400 ng/m3 emphasizing again the likely

importance of LNOx in the production of nitrate aerosols.

Sulfate concentrations are much higher (100-200 ng/m3 STP) for air parcels initialized near 16-

17 km for ZF2 and ZF3 likely indicating stratospheric sources while air parcels near 14-15 km

show levels below 100 ng/m3 STP. We note that sulfate along the trajectories influenced by

Chinese pollution during ZF2 increases by 60%, approximately 50h before our measurements

which could indicate the formation of sulfate aerosol from SO2. It has previously been reported

that sulfate has a lifetime of a few days in the troposphere (Hidy and Blanchard, 2016). The

rather short lifetime of sulfate is due to absorption in precipitation, or solubility (Hidy,1973).

The global mean residence time of tropospheric sulfate against dry and wet deposition is about a

few days (e.g., Park et al., 2004).

The GEOS-Chem model showed higher sulfate levels than the results from IC due to relatively

weak scavenging of $SO_2$ and/or $SO_4^{2-}$.

The aircraft field campaigns of the StratoClim project were held in July and August 2017 at the

Tribhuvan International airport (KTM; 27.70°N, 85.36°E, Katmandu-Nepal). In-situ aerosol

measurements within the AMA were carried out using the aerosol mass spectrometer, ERICA.

Flights KTM 01, and 02 held on 27[th] and 29[th] July showed a low level of sulfate at 360°K

(potential temperature) corresponding to an altitude of 15km. The sulfate concentration was

almost equal to zero on 10[th] August during flight KTM 08 at 16km and 370°K (Stephan

Borrmann-4[th] ACAM workshop 28-06-2019). The very low levels of sulfate sometimes observed

in the StratoClim campaign near 360°K-380°K are consistent with our IC analysis results of

sulfate ionic concentration during flight ZF2 held on 15[th] August at the same altitude and

potential temperature.

*9. Summary and Conclusions*

The chemical composition of the ATAL has been investigated using offline IC analysis of aerosol impacted samples collected onboard the zero-pressure balloon flights as part of the BATAL campaigns. The measurements of the 2017 summer campaign indicate the dominating presence of nitrate and nitrite aerosols with concentrations between 88 and 374 ng/m$^3$ STP. Our first flight (ZF2) on 15$^{th}$ August 2017, occurred within the AMA and thus sampled air masses therein. In situ measurements revealed the presence of $NO_3^-$, and $NO_2^-$ aerosols (60-200 ng/m$^3$ STP) of sizes ranging between 0.05-2 µm. The second flight (ZF3) on 21$^{st}$ August 2017, however, occurred at the edge of the anticyclone and subsequent in situ measurements revealed the presence of larger particle size $NO_3^-$ and $NO_2^-$ aerosols at higher concentrations (87.3-343 ng/m$^3$ STP). Throughout the flights during the 2017 summer campaign, sulfate aerosol remained below the detection limit of the system (10 ng/m$^3$ STP) much lower than the results from the GEOS-Chem model simulation (80-120 ng/m$^3$ STP). The higher model sulfate levels than that from IC are believed to be due to relatively weak scavenging of $SO_2$ and/or $SO_4^{2-}$ in the model. Unlike the summer, $Ca^{2+}$ and $SO_4^{2-}$ were found on all four stages in sizes ranging between 0.2-0.05 µm together with traces of $NH_4+$ which couldn't be quantified in the winter campaign. The winter flight sampled residuals from the 2017 Canadian wildfires which affected stratospheric aerosol loadings for several months.

We study the influence of convection on those measurements using back trajectory calculations collocated with geostationary satellite observations. We show that ZF2 and ZF3 were influenced by convection over Western China, the Bay of Bengal as well as Myanmar, Thailand, and Laos. The model was able to reproduce the convective transport from the mid-troposphere (9-12 km) to the upper troposphere (14-15 km). There was no indication of the transport of these air parcels from the boundary layer. Although HIMAWARI-8 observations showed the convective transport reproduced in MERRA-2, the mixture between horizontal and vertical transport wasn't visible in trajectory calculations. Tropical convection could explain the rapid ascent of the air parcels to higher altitudes since other mechanisms namely, radiative heating would delay the transport of air parcels from the middle to the upper troposphere.  While the model seems to represent convection

in the upper troposphere (14-15 km) with the rapid ascension of air parcels, the model's ability to simulate convective influence at higher altitudes seems to be limited.

We used the GEOS-chem model simulations with and without lightning $NO_x$ emissions to understand the contribution of lightning to nitrate aerosol. The flights, ZF2 (Aug. 15[th]) and ZF3 (Aug. 21[st]) occurred during a period where the levels of nitrate were relatively small ($<$100 ng/m$^3$ STP), with minimal influence of lightning $NO_x$ in contrast with other periods largely affected by nitrate produced by LNOx. As shown by trajectory calculations in Fig. 5, flights ZF2, and ZF3 sampled air masses localized at the border of the Asian anticyclone. Fairlie et al., 2019 showed that the eastern part of the ATAL anticyclone depicts a peak of ammonium contribution from Chinese emissions. The western core of the ATAL on the other hand is seen to be enriched with 80% of anthropogenic sources from India with the southern and eastern flanks of the anticyclone showing peaks of Chinese contribution wherein nitrate concentrations were found to be the highest.

Since the ASM varies in spatial dimensions and methodology, inconsistencies in the seasonal and interannual contribution to the ATAL are expected. Mineral dust is considered to be the most abundant type in the troposphere, its main emission source being from arid, and semi-arid regions (Huneeus et al., 2011). $CaCO_3$ is considered to be one of the most important components of mineral dust, of which about 1.3 Tg of $CaCO_3$ is loaded in the troposphere (Scanza et al., 2015). During atmospheric transport heterogeneous reactions occur with trace gases thereby forming more soluble species resulting in the increased CCN (cloud condensation nuclei) activity of mineral dust particles. Flight, ZF2 sampled air masses within a cloud showing the presence of $Ca^{2+}$ and $NO^{3-}$ on the same stage ($< 0.15\mu$m size particles). This implies the formation of $Ca(NO3)_2$ on the reaction of $CaCO_3$ with $HNO_3$.

Indeed, the atmosphere is an amalgamated den in which gaseous species, particulates, and liquid droplets co-exist at the same time. Through our balloon campaigns during the ASM with simultaneous offline measurements of inorganic species and thereby comparing the results with model simulations, we were able to understand if not fully answer the many unanswered questions on the existence & behavioral pattern of these ionic species of interest. We will continue to research this area with improved techniques & additional experimentation.

*Data availability*. We plan to keep those data on the Langley Archive database together with the model results

*Author contributions*. HV led the preparation of the paper. AP and NR contributed to the chemical analysis of the balloon samples. MVR, HG, JPV, SK, AKP, and GB organized the balloon flights. FW contributed to the analysis of the COBALD data. HL and BZ performed GEOS-Chem model simulations and assisted with model output analysis. KB analyzed HIMAWARI-8

*Competing interests*. The authors declare that they have no conflict of interest.

*Acknowledgement.* HL, KMB, BZ, JPV acknowledge funding support from the NASA Atmospheric Composition Modeling and Analysis Program (ACMAP) and the Upper Atmospheric Research Program (UARP). NASA Center for Computational Sciences (NCCS) provided supercomputing resources. The GEOS-Chem model is managed by the Atmospheric Chemistry Modeling Group at Harvard University with support from NASA ACMAP and MAP programs.  The author acknowledges Duncan T. Fairlie for his contribution to this effort.

*Financial support.* This research has been supported by the NASA Atmospheric Composition Modeling and Analysis Program (ACMAP), and by the ANR (Agence Nationale de La Recherche) under grant ANR-10-LABX-100-01 (French LabexVOLTAIRE managed by the University of Orleans).

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

1    **Figures.**

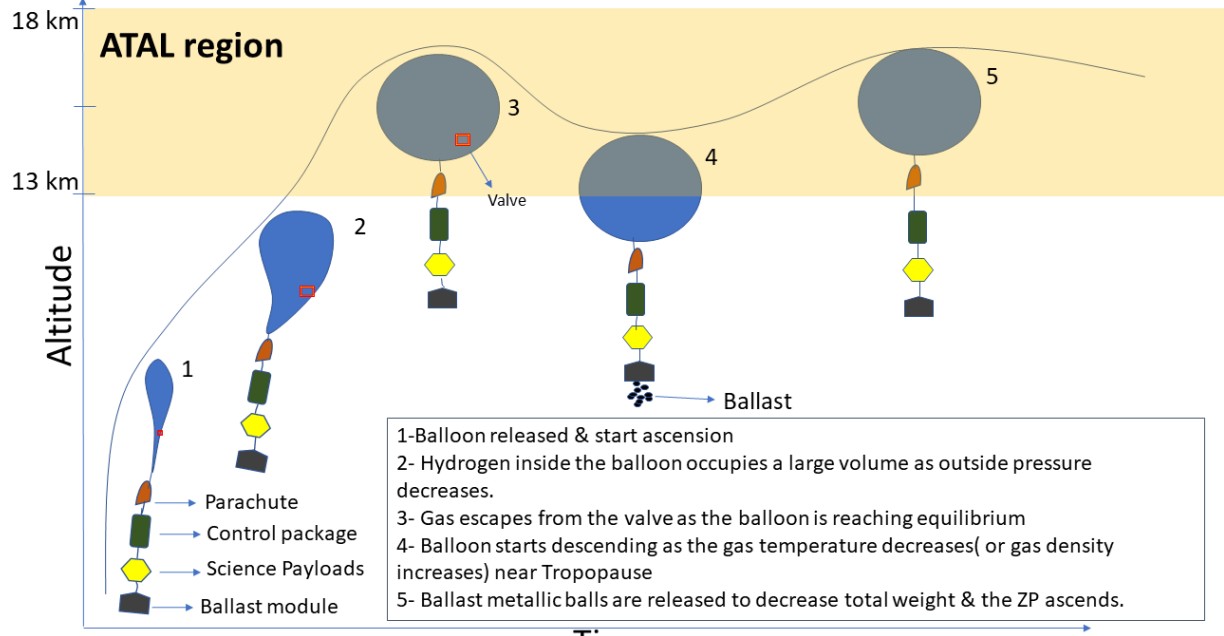

1-Balloon released & start ascension
2- Hydrogen inside the balloon occupies a large volume as outside pressure decreases.
3- Gas escapes from the valve as the balloon is reaching equilibrium
4- Balloon starts descending as the gas temperature decreases( or gas density increases) near Tropopause
5- Ballast metallic balls are released to decrease total weight & the ZP ascends.

3
4
5

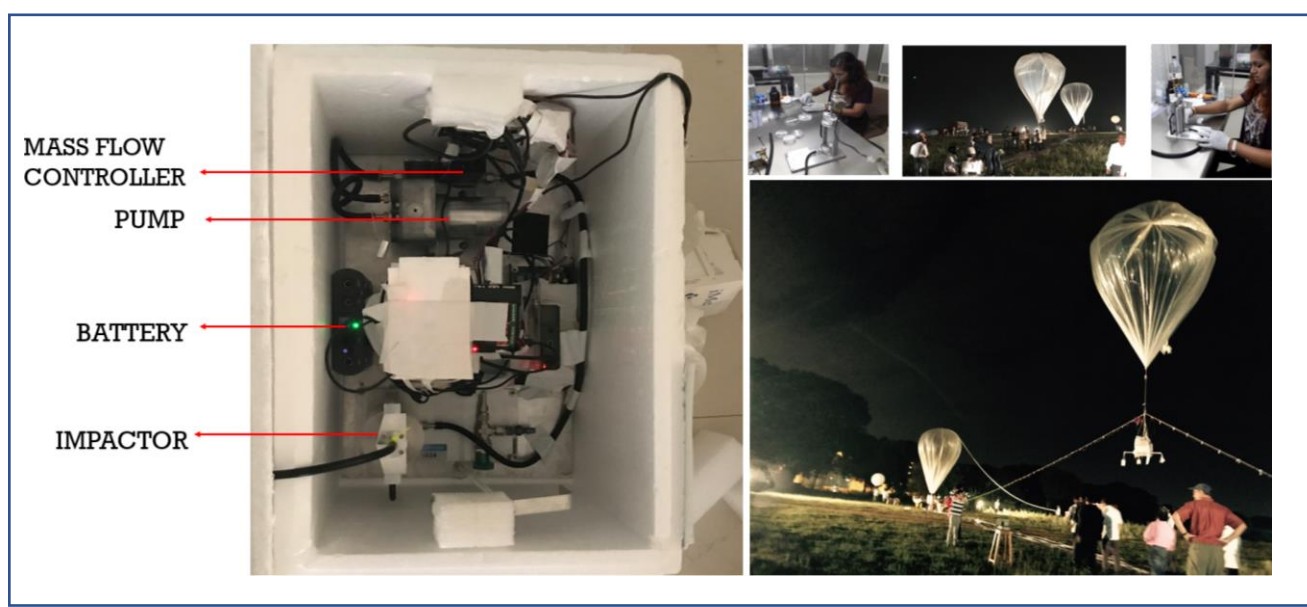

8
9
10
11

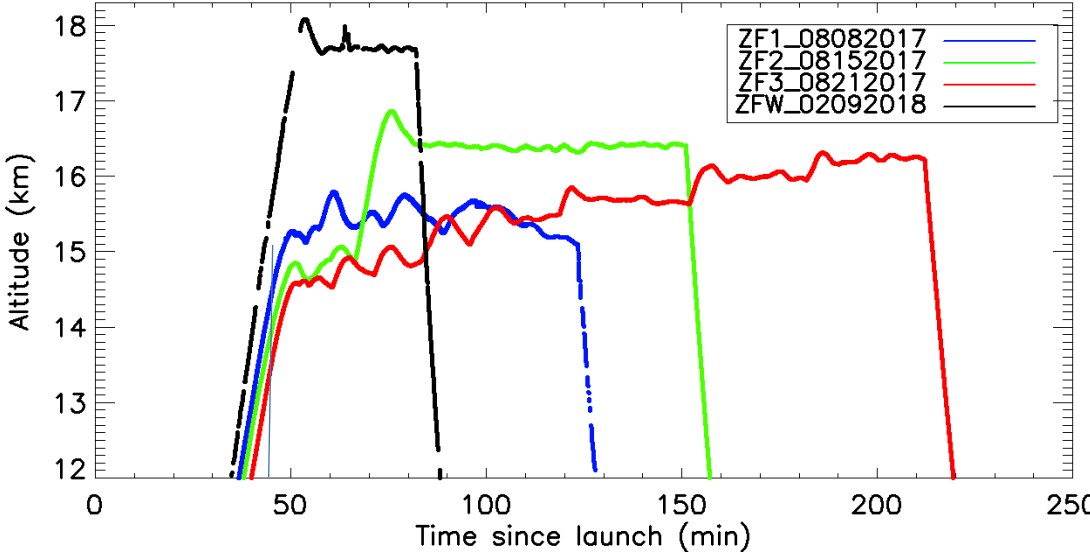

Figure 1. (Top). Schematic diagram of the zero-pressure flight concept. (Middle) Picture of the
science payload, impactor preparation, and balloon flight launch (Bottom) Time-height curves of
the GPS altitudes of the 3 zero-pressure flights during summer 2017, in comparison with that of
winter 2018, launched from TIFR-BF, Hyderabad, India.

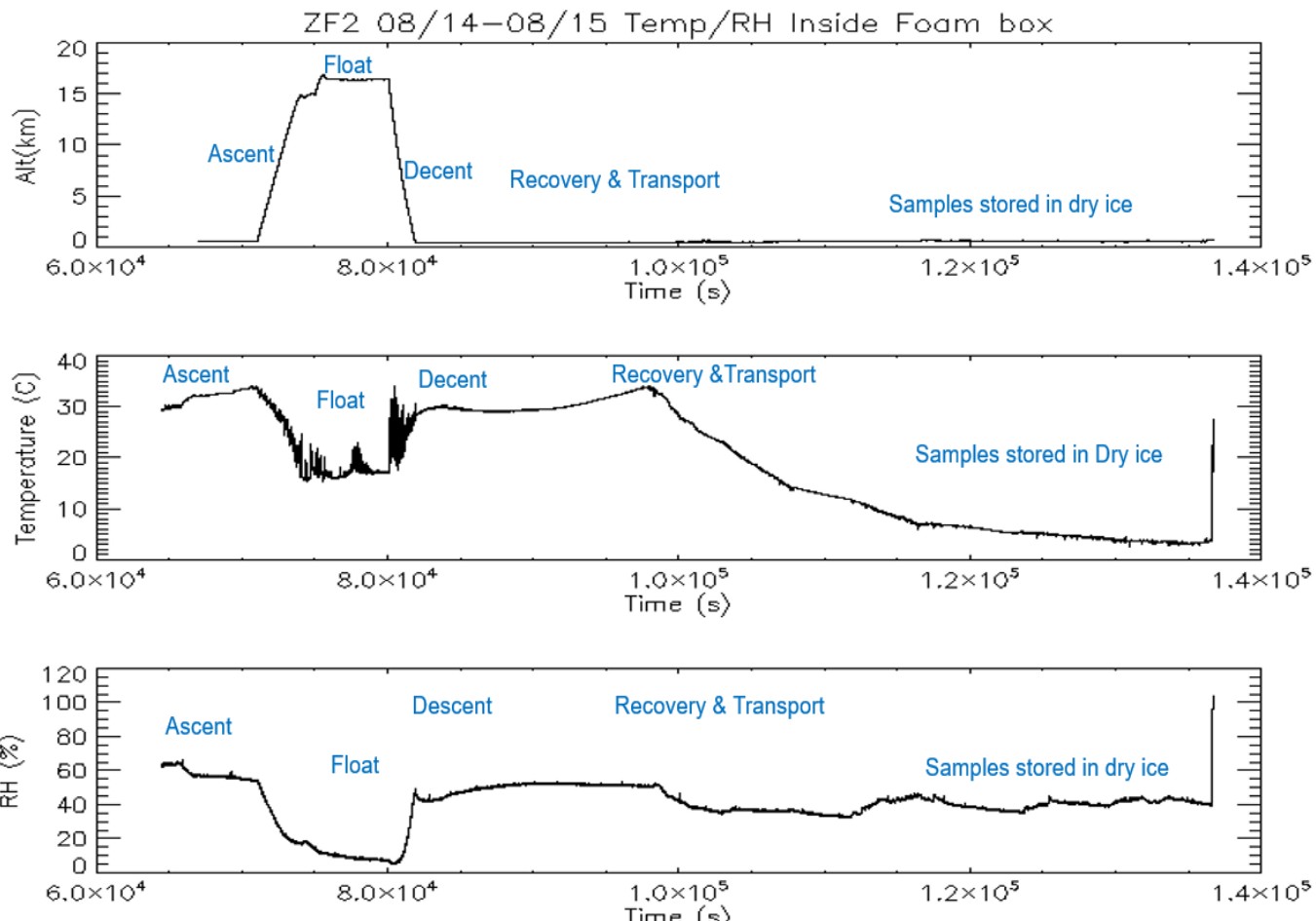

2    Figure 2. Time series of the Altitude, Temperature, and Relative humidity profiles of the samples

3    inside the foam box during the ZF2 flight.

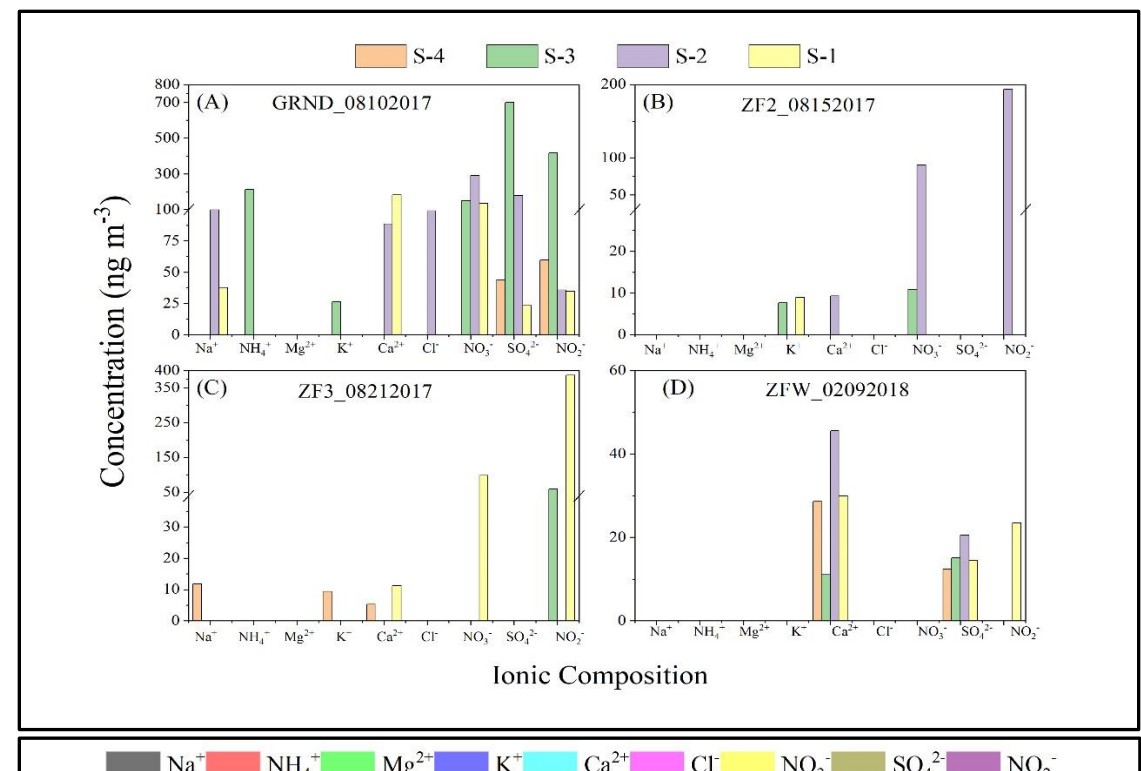

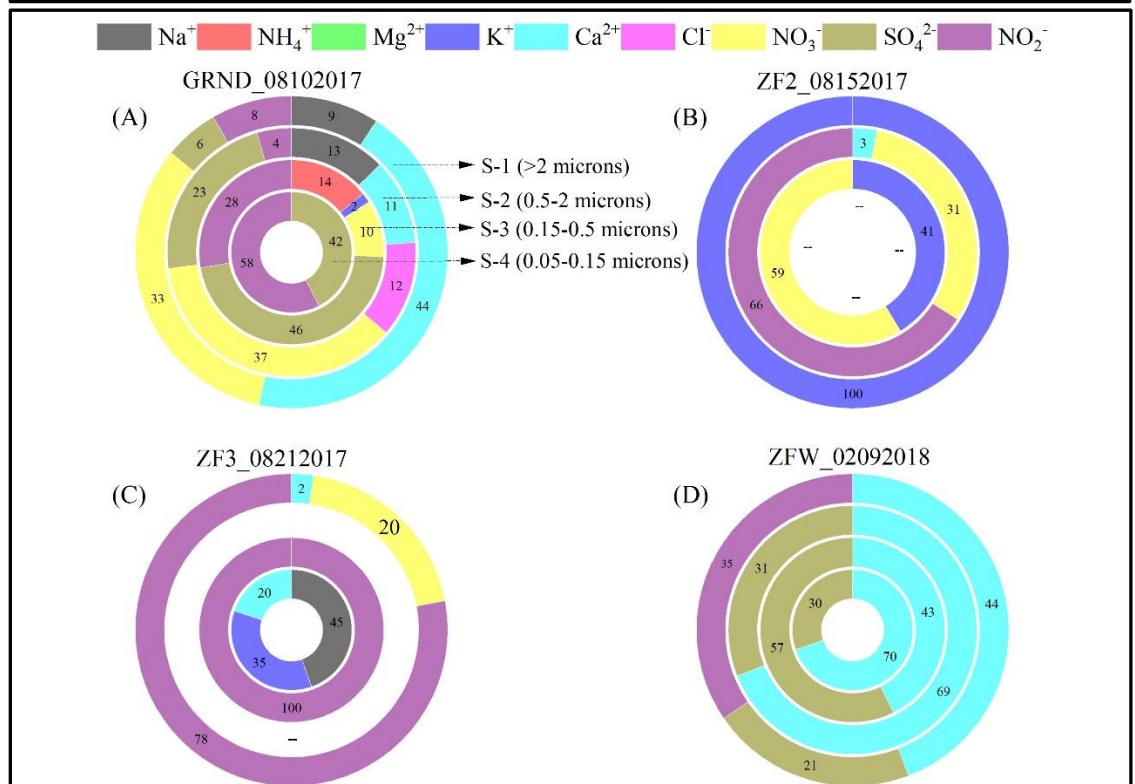

Figure 3. Results from the analysis of inorganic aerosol. (Top) Aerosol ionic composition of the filters collected on (A) the ground (B) ZF2 (C) ZF3 in Summer 2017, and (D) ZFW in Winter 2018. (Bottom) Percentage distribution of individual ions. S1 to S4 indicate the four stages of the impactor. The size cut off is > 2, 0.5, 0.15, and 0.05 micron for S1, S2, S3, and S4, respectively.

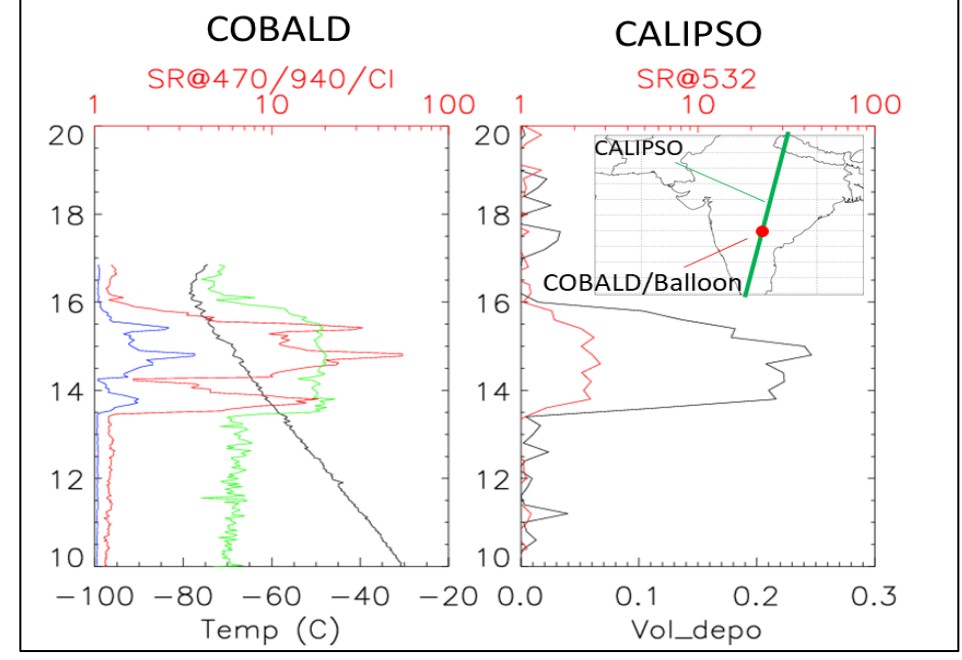

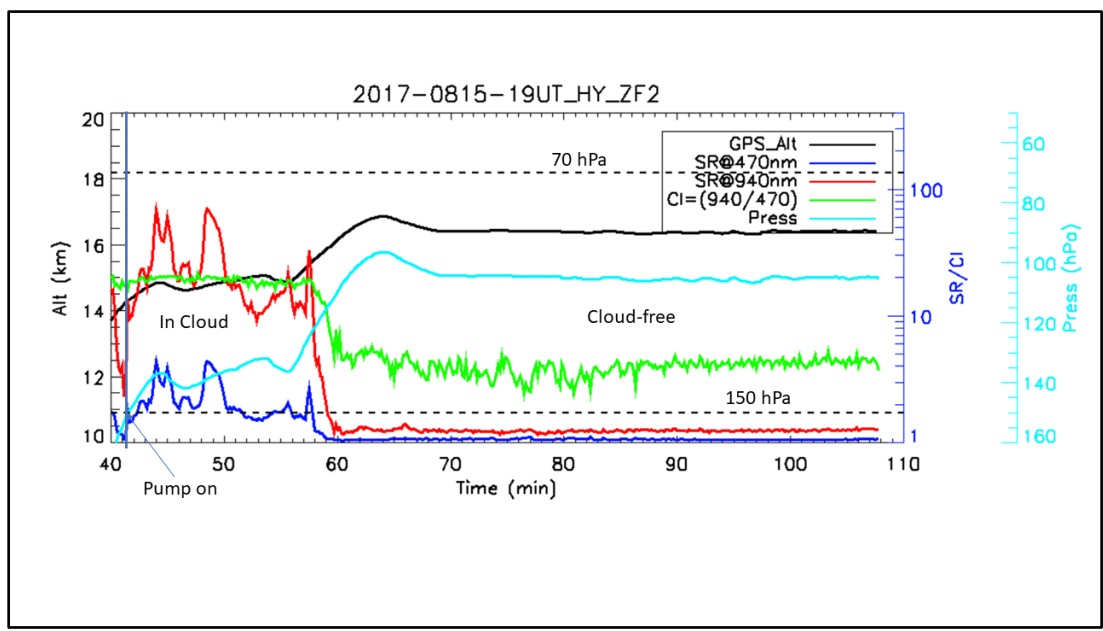

Figure 4. (Top) COBALD balloon *In Situ* and CALIOP Satellite Scattering Ratios (SR), volume depolarization and Color Index (CI) profiles collocated in time and space (within 20 km and 1 h) on August, 15th at 19 UT. (Bottom) Time series along ZF2 of Scattering Ratios (SR) at 940 nm and 470 nm from COBALD and GPS altitude (colored with ascent rate) and measured pressure from the Imet radiosonde.

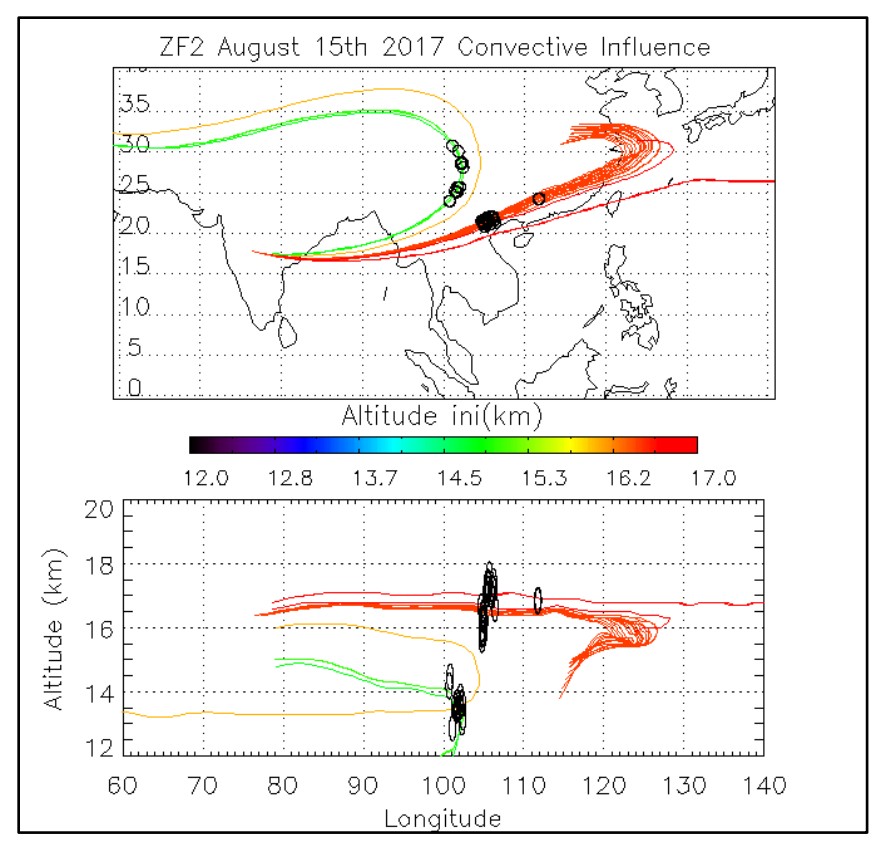

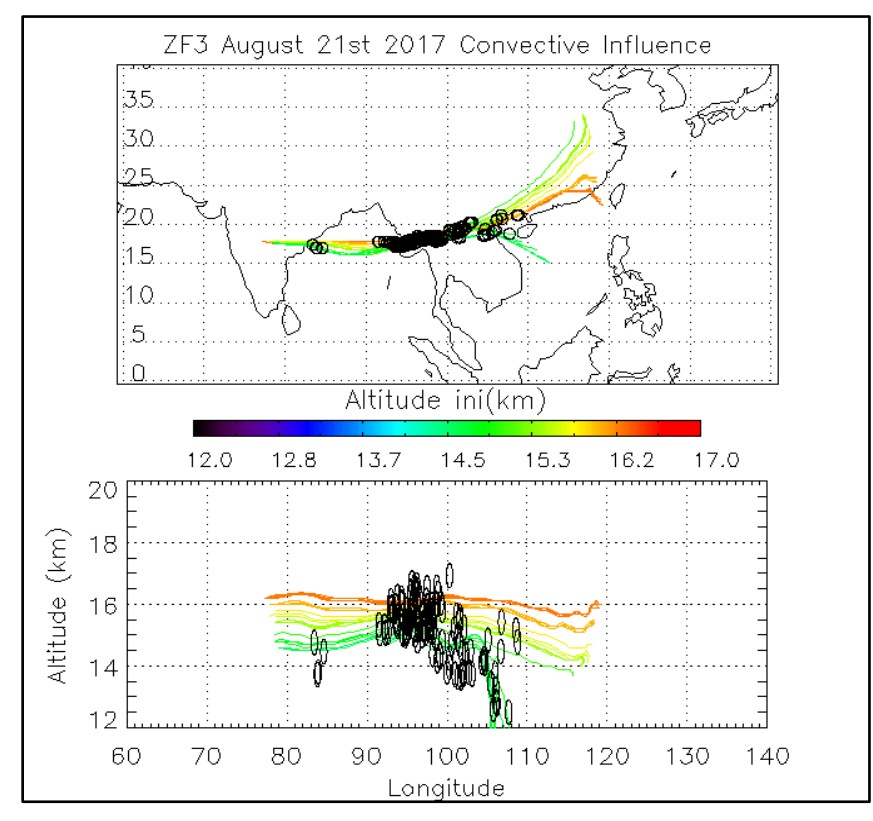

1 Figure 5. Back-trajectories initialized from ZF2 (08/15) and ZF3 (08/21) measurements between

2 150 hPa and 70 hPa. Black dots along the trajectories are the position of convective systems

3 intersecting air masses sampled during the balloon flight.

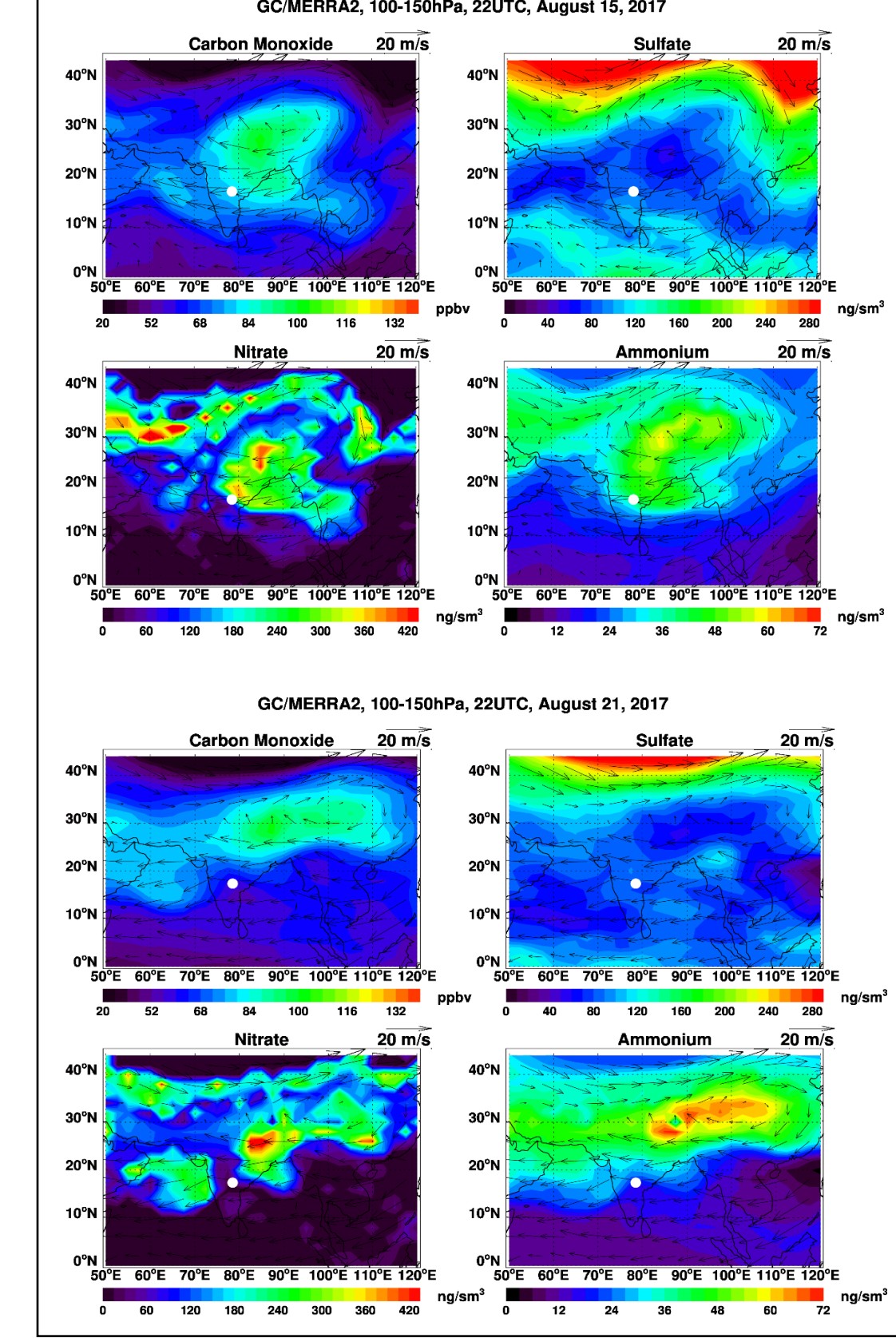

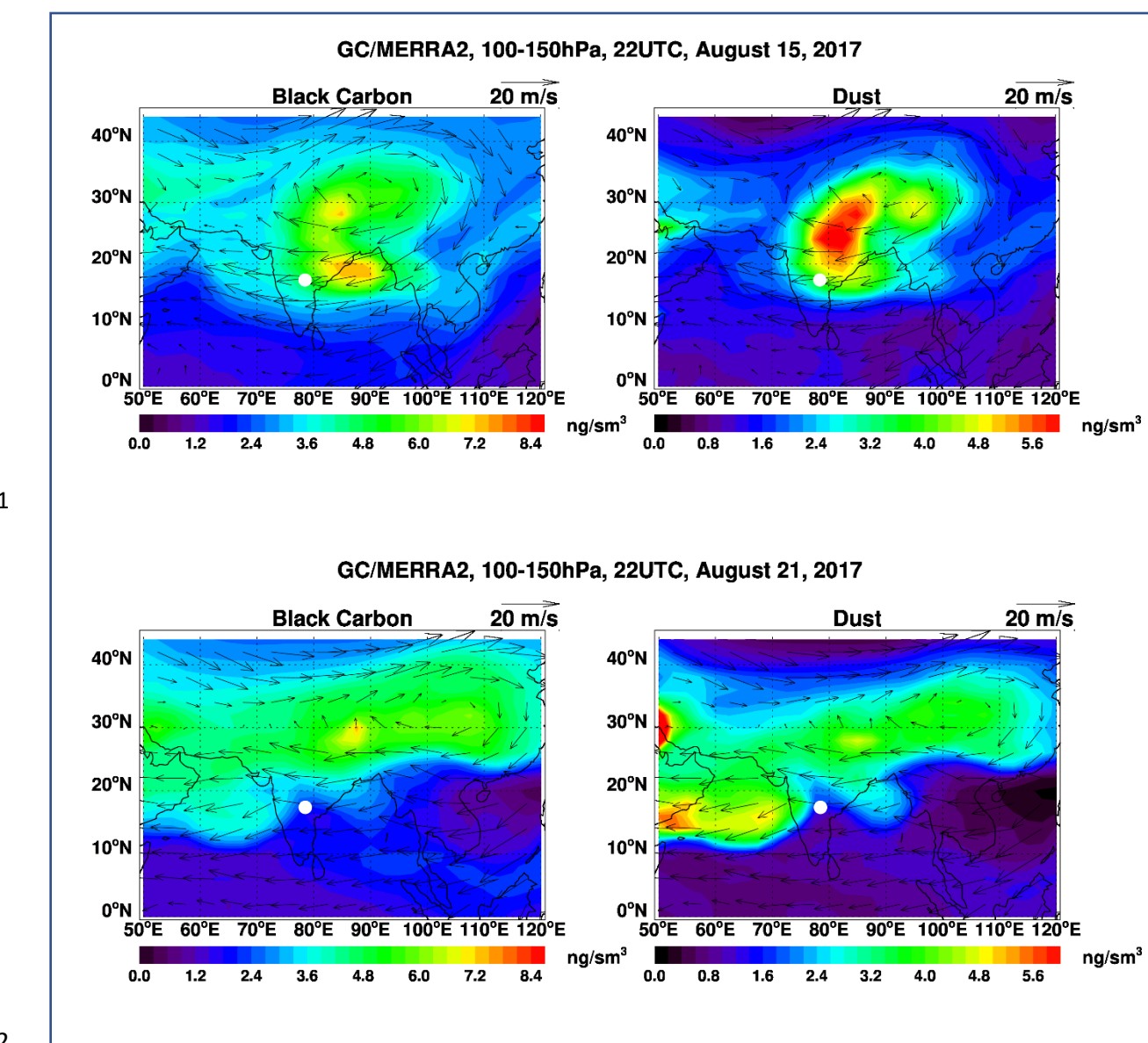

Figure 6. (Above) GEOS-Chem model-simulated carbon monoxide (CO, ppbv), sulfate ($SO_4^{2-}$, ng/m$^3$ STP), nitrate ($NO_3^-$, ng/m$^3$ STP), and ammonium ($NH_4^+$, ng/m$^3$ STP) (Top panels). (Below) GEOS-Chem model-simulated black carbon (BC, ng/m$^3$STP), and dust ($Ca^{2+}$, ng/m$^3$STP) concentrations averaged over 100-150hPa at 22UTC, August 15[th], and August 21[st] 2017 (Bottom panels). Standard temperature and pressure (STP) is 298K and 1013.25 hPa, respectively. Arrows denote wind direction while the white circle indicates sampling location, Hyderabad, India.

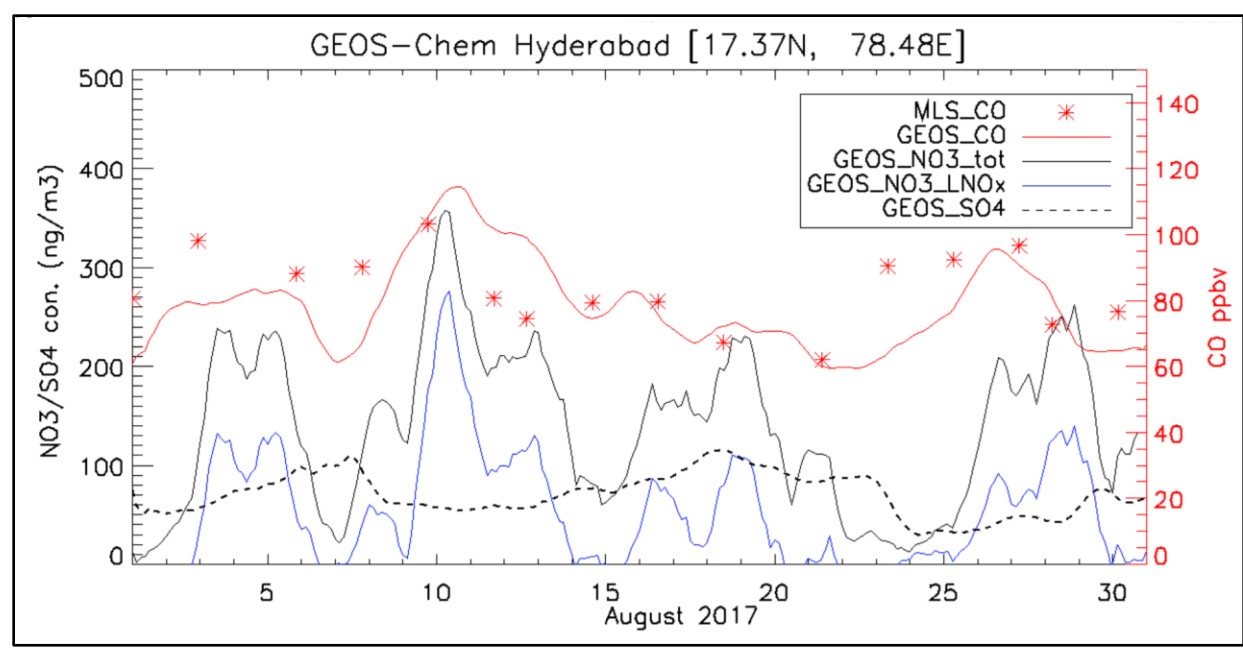

Figure 7. Time series of simulated 3-hourly CO, $SO_4^{2-}$, and $NO_3^-$ concentrations averaged over 100-150hPa at Hyderabad during the ZF2 and ZF3 flights on 15[th] Aug. & 21[st] Aug. 2017. Also shown are concentrations of $NO_3^-$ due to lightning NOx emissions (NO3_LNOx). See text for details.

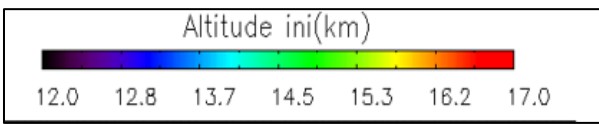

Figure. 8. GEOS-Chem model-simulated $NO_3^-$, CO, and $SO_4^{2-}$ STP concentrations extracted

along the trajectory lines during flights ZF2 and ZF3 (Fig. 5).

4 # SUPPLEMENTARY FIGURES:

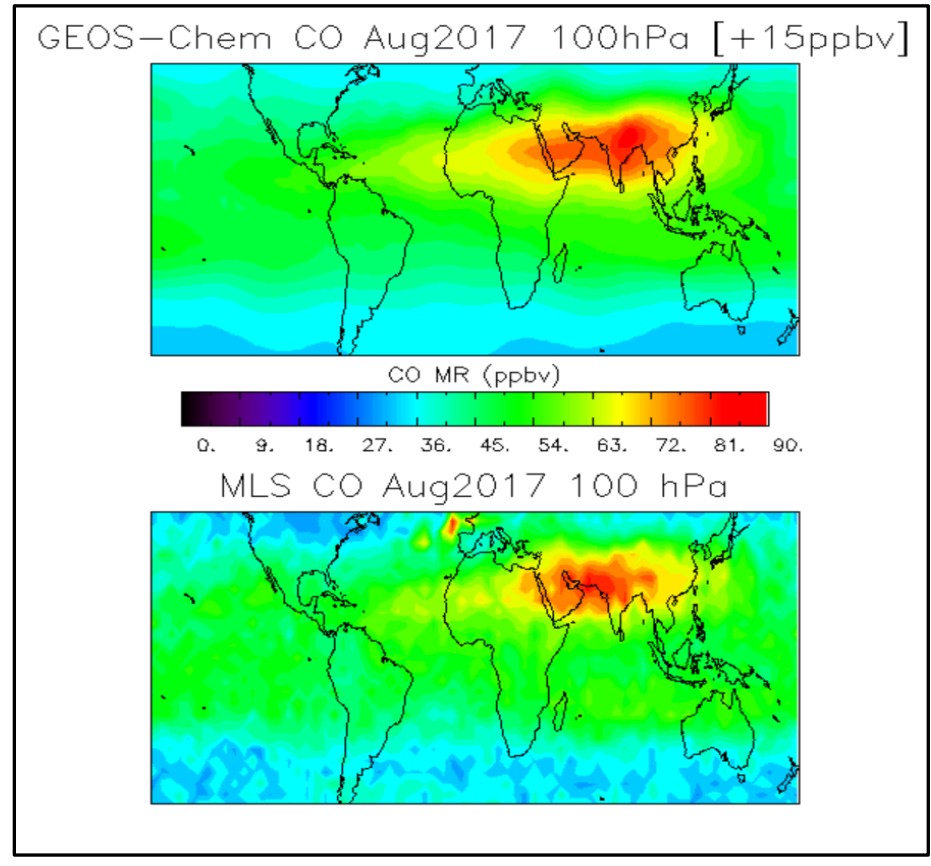

Fig. S1 Maps of Carbon monoxide (CO) for GEOS-Chem (above) and Microwave Limb Sounder
(below) at 100 hPa for August 2017. An offset of +15ppbv is added to GEOS-Chem to make the
comparison with MLS easier. The general patterns between MLS and GEOS-Chem are very
similar with a maximum of CO associated with the Summer Asian Monsoon extending up to the
Arabic Peninsula. However, the maximum of CO simulated by GOES-Chem is located over
Eastern India while MLS maximum is shifted to Western China and Pakistan.

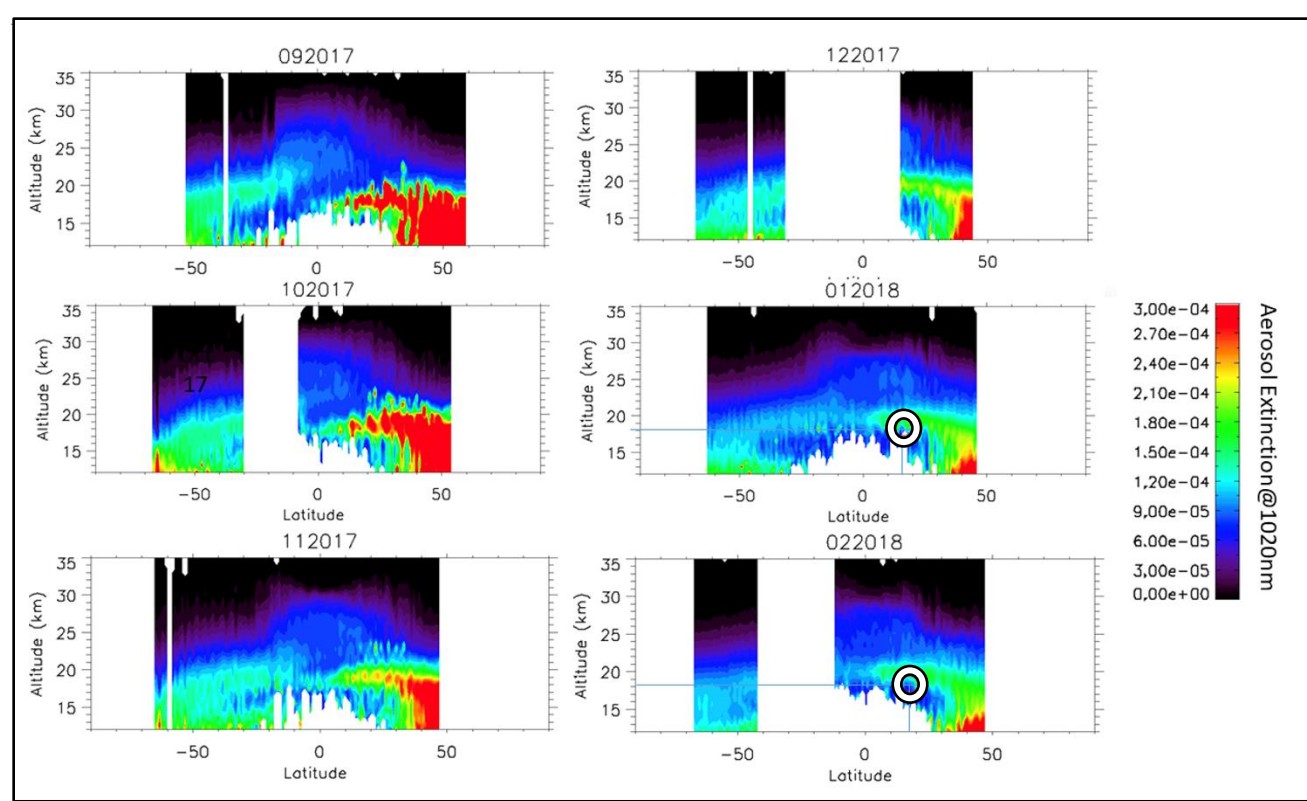

Figure S2. Zonal mean aerosol extinction at 1020 nm derived from the SAGE III/ISS V051data
products between September 2017 and February 2018. Ice clouds in the troposphere have been
removed using a threshold of color ratio (521nm/1020nm) below 2 (Vernier et al., 2015).
Increase of aerosol extinction between 10-50°N and 13-21 km is observed from September 2017
to the end of 2017 as a result of the Pacific Northwest Canadian PyroCbs which injected smoke
in the Upper Troposphere and Lower Stratosphere in August 2017. A residual of the smoke
plume is still detected up to February 2018. The white rings show the location of the balloon
flight at the bottom of the aged smoke plume.

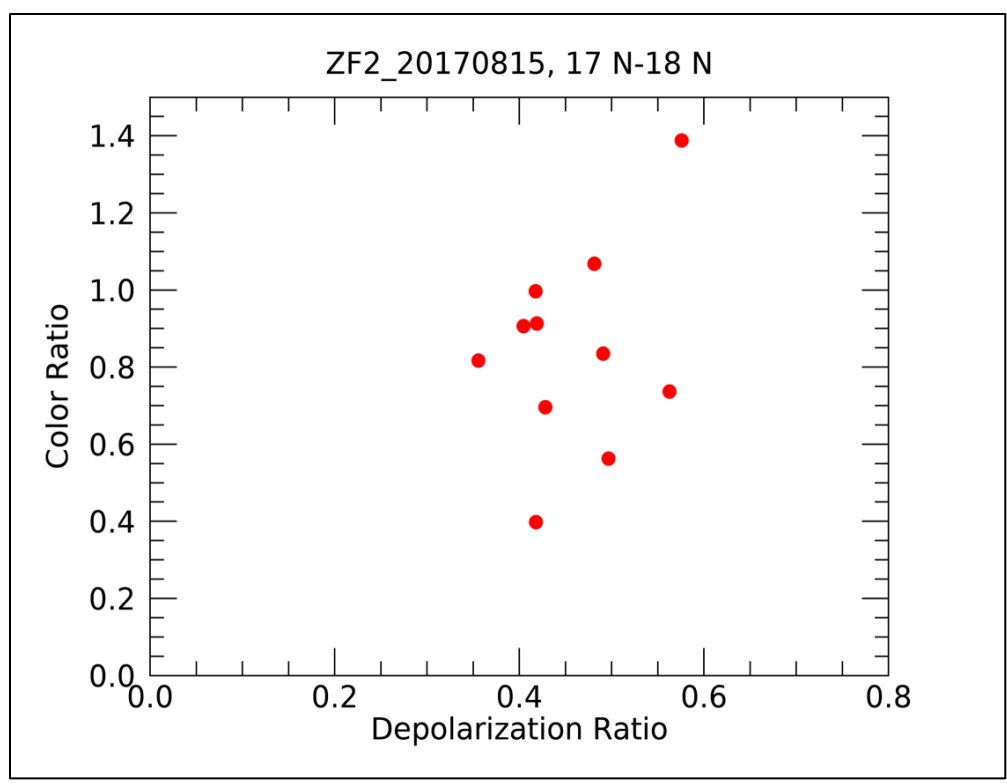

Figure S3. Cirrus cloud layer properties using CALIOP L2V4.2 Cloud Layer product for August
15[th] 2017 between 17.12°N and 17.92°N corresponding to the profiles shown in Fig.4.
Depolarization ratio versus color ratio plot for these layers which indicates the presence of
aspherical large particles consistent with the properties of sub-visible cirrus clouds (mean
AOD~0.03+/-0.02).