# Peer review of "Exploring the inorganic composition of the Asian Tropopause 1 Aerosol Layer using medium-duration balloon flights 2 3 4 5 6 Hazel Vernier1, Neeraj Rastogi2, Hongyu Liu3,4, Amit Kumar Pandit3, Kris Bedka4, Anil Patel2, 7 Madineni Venka"

_Atmospheric Chemistry and Physics, 2021_

## Author Response (AR1)

We thank the two reviewers for their valuable suggestions and constructive criticism, which helped improve our analysis and manuscript. Below we provide a detailed point-by-point response (in Blue) to the reviewers' comments (in Black).

RESPONSE TO REVIEWER 1

RC1.1 The authors mention that both nitrate and nitrite were important constituents of the aerosol composition; however, they only really focus on the potential sources of nitrate. Nitrite is a generally unusual component of aerosol composition and I would expect to have different physicochemical properties than nitrate, including in the ATAL and its impact on cirrus clouds and radiative properties. A discussion on nitrite, including sources and what form it is in the ATAL would improve the paper and make it more appropriate for a research article than a measurement report.

Authors:  We agree with the reviewer's comment. A section on nitrite sources and previous findings has been added to the revised manuscript (Section 7, Page 12).

RC1.2 The authors mention that the GEOS-Chem predictions have much higher sulfate than the filter measurements and provide one short sentence speculation about why without further discussion or ramifications in a model having too high sulfate compared to observations. Ramifications of too high sulfate should be addressed, as that is an important finding. Also, how does the sulfate observations here compare to prior studies?

Authors: - The model simulates sulfate-nitrate-ammonium aerosol thermodynamics coupled to ozone-NOx-hydrocarbon-aerosol chemistry (Park et al., 2004). The GEOS-Chem model showed higher sulfate levels than the results from IC likely due to relatively weak scavenging of $SO_2$ and/or $SO_4^{2-}$. The very low levels of sulfate sometimes observed in the StratoClim (Borman et al., 2019, ACAM workshop campaign) near 360°K-380°K are consistent with our IC analysis results of sulfate ionic concentration near the same altitudes and corresponding potential temperature.

For more details please refer to the revised MS, section. 8 (P 13, L 21; P 16, L 1 onwards).

In addition, the fact that we found sulfate during the 2018 winter campaign demonstrates that the sampling technique can be used to detect sulfate above 10ng/m3. Thus, the lack of sulfate found in the summer 2017 indicates its absence rather than a limitation of the sampling/analysis techniques.

RC1.3: - The authors use GEOS-Chem as is without any verification if the model predictions match observations (e.g., is CO at the right location / concentration). Satellite retrieved observations of CO could help or even be used instead of GEOS-Chem CO to make this point.

Further, other satellite products could be used to try to make the points the authors are trying to make here. On top, it is unclear how well GEOS-Chem is performing for nitrate compared to observations. One obstacle behind this comparison is (a) was ISORROPIA used to predict the thermodynamic partitioning of nitrate with aerosol phase and (b) what nitrate was used within ISORROPIA. If, for example, GEOS-Chem only has ammonium nitrate and not nitric acid trihydrate (NAT) or more refractory aerosol (sodium or calcium nitrate), then the discussion about nitrate with the filter measurements does not work here.

Authors: We have now analyzed MLS CO and added the results in the paper (Fig.9, Fig. S2). It shows a relatively good agreement with GEOS-Chem simulation for August 2017. In addition, Simulation with an earlier version of GEOS-Chem showed that the model simulates elevated CO in the upper troposphere over the Asian summer monsoon region, resembling the observations of CO from the EOS Microwave Limb Sounder (MLS; Li et al., 2005). The elevated CO results from the transport of boundary layer pollution by monsoon convection and orographic lifting to the upper troposphere over South Asia. To our knowledge, the model has not been evaluated with in situ nitrate observations, if any. ISORROPIA II is a thermodynamic equilibrium model for the $K^+$–$Ca2^+$–$Mg2^+$–$NH^+ 4$ – $Na^+$–$SO2^- 4$ –$NO^- 3$ –$Cl^-$–$H2O$ aerosol system which includes ammonium, sodium and calcium nitrate. However, the model does not include the formation of NAT. We have modified the text accordingly to explain the limitation of GEOS-Chem for NAT formation. We have added in section 8 additional information on ISORROPIA-II

RC1.4: - The nitrate being either NAT or potentially refractory aerosol and not ammonium nitrate is an important finding, especially in regards to the physicochemical properties of aerosol in ATAL. However, this is not explored and expanded upon. Also, could this be a measurement artifact (see next comment)? The fact that prior studies observed ammonium nitrate and not the aerosol reported here needs to be further explored as the findings here are very different.

Authors: -

We haven't made any conclusive statements on this particular aspect. There are possible inferences from the data and literature. As reported, concentrations of $NO^{2-}$ and $NO^{3-}$ were relatively very high (87-343 ng/m3) compared to any other measurable ions. The detection limit of anions was about 5 ng/m3 and cations was about 10 ng/m3. If the NO3- existed as NH4NO3, the expected NH4+ would have been much higher than its detection limit, which was not the case. Further, a high concentration of NO2- was also observed, which could be the intermediate product of NAT particles. These observations allowed us to infer the observed NO3- as possible NAT. We shall be conducting more balloon flights to make conclusive statements in near future. In addition, since ammonium sulfate is relatively unstable, we believe that it would have disappeared when the samples were brought back to the ground. A discussion of instability of ammonium nitrate is provided (P 9, L2).

« The dissociation of $NH_4NO_3$ into gas-phase $HNO_3$ and $NH_3$ increases sharply with increasing temperature and relative humidity (Seinfeld et al., 1982; Lightstone et al., 2000), leading to a significant loss of particulate nitrate (PN)."

In addition, the measurement locations between StratoClim and the BATAL campaign in Hyderabad could also explain why the ATAL composition results might be different. The StratoClim flights were located near the center of the AMA while the BATAL flights took place at the edge. Fairlie et al. (2020) showed that Indian pollution is likely to influence air masses within the AMA while Chinese pollution forms a "horse shoe" shape on the East side of the anticyclone.

RC1.5: - The authors make a speculation that Canadian wildfires impacted the ATAL in the abstract and the conclusion. With it being in the abstract, this would lead readers to expect more attention on this detail in the paper instead of just a brief passing sentence in the conclusion. This could be a section in of itself, especially regarding how forest fires could impact ATAL. Currently being in just abstract and conclusion further makes this paper feel more like a measurement report instead of a research article.

Authors: - We thank the Reviewer for pointing this out. The section on Canadian wildfires is discussed in the revised manuscript (Section 5, P 9) and a plot of aerosol extinction from SAGE III/ISS is added in the supplementary material to demonstrate that the sampling took place in the bottom part of aged smoke layer.

RC1: - Page 8, line 26 – 31 and page 9, line 1 – 6, the authors discuss how ammonium nitrate could be lost on filters. However, there was no discussion if experiments were conducted to determine how much loss occurred. Further, the authors mention NAT (page 8, line 15) could explain the lack of charge balance in the observations they show. But, there is no discussion about how stable NAT would be on filters prior to and during the freezing of the filters and during the preparation of the filters for sampling. Further discussion / exploration of this is needed to put the paper into context of prior studies and for use in comparisons against chemical transport models.

Authors:

The Reviewer has raised an important point. The NAT particles are stable under stratospheric conditions. In the process of sampling, transport, and extraction, there is a finite possibility of NAT particle losses, if they were collected. However, if the NO3- was present in another form (refractory nitrate then they shall remain relatively stable during the said processes). Observed cations were close to or below the detection limit compared to the significant concentrations of NO3- and NO2-. This observation along with the higher abundance of NO2- allowed us to infer the possible presence of NAT particles. However, we agree with the Reviewer that if it was NAT then the reported concentrations be considered as the lower limit, presuming some losses (unquantifiable) during the sampling, transport, and extraction process. We have added this information in the revised manuscript (P 8, L 27).

RC1: - Page 9, line 17 – 18, the authors quickly mention the pump was on for 16 minutes. This is significantly shorter than the 2 hours the authors said was needed to collect enough aerosol to have measurements above detection limit. This needs to be clarified.

Authors: -

Flight ZF2 floated for almost 2 hrs at an altitude between 16-17 km. It was during this flight that ZF2 sampled through a thin ice cloud for 16 minutes while it sampled for more than an hour at the same altitude in a cloud-free region. Please refer to the revised manuscript (P 9, L 25).

RC1: - The description of GEOS-Chem needs to be moved to methods, and further description of GEOS-Chem needs to be provided—was ISORROPIA used, what forms of nitrate were included, was nitrite in the model, how was SOA modeled, etc.

Authors: -
 The description of GEOS-Chem is now moved to section 2.6. A short section describing the NASA Langley Trajectory Model is also added (section 2.5). Further description of GEOS-Chem has been provided, as suggested: "The model simulates black carbon (Park et al., 2003), primary and secondary organic aerosols (SOA; Pye et al., 2010), sulfate-nitrate-ammonium aerosol thermodynamics coupled to ozone-NOx-hydrocarbon-aerosol chemistry (Park et al., 2004), mineral dust (Fairlie et al., 2007; Ridley et al., 2014), and sea salt (Jaegle et al., 2011), treated as an external mixture. SOA uses the volatility-based scheme (VBS) of Pye et al. (2010). Sulfate-nitrate-ammonium thermodynamics is computed using the ISORROPIA-II thermodynamic equilibrium model of Fountoukis & Nenes (2007). Aerosol wet deposition includes rainout and washout due to large-scale precipitation as well as scavenging in convective updrafts (Liu et al., 2001). Scavenging of aerosol by snow and mixed precipitation is described by Wang et al. (2011, 2014). Dry deposition of dust and sea-salt aerosols uses the size-dependent scheme of Zhang et al. (2001). Dry deposition for other aerosols follows the resistance-in-series scheme of Wesely (1989).

RC1: - There were four flights of the payload, yet only three flights are shown. Why is one flight not included in Fig. 3 or any discussion?

Authors: -

Of the four flights, ZF1 was a test flight to understand and maintain the float altitude using ballast. The same has been clarified in the revised manuscript (P 6, L 28).

RC1: - Page 10, line 1, the authors have the description of trajectories and deep convection different than in Fig. 5.

Authors: -

Agreed and corrected (Pg10, Line 22).

RC1: - Page 11, line 16, make sure Chem in GEOS-Chem is capitalized.

Authors: -

 Corrected (Pg12, Line 27).

RC1: - Page 11, line 18 – 20, the authors make a statement about upper troposphere (UT) NOx lifetime and its sinks. However, the normal thought about UT NOx lifetime is that it is about 2 days. Where did this shorter lifetime and sinks come from?

Authors: -

The lifetime of $NO_x$ is approximately 3h in the region of the outflow of thunderstorms due to the production of methyl proxy nitrate and alkyl, and multifunctional nitrates (Nault et al., 2017).

RC1: - Page 12, line 4 – 8, the authors use mass concentration for CO instead of volume mixing ratio; however, in the figures, CO is reported in volume mixing ratios. Please clarify which is correct.

Authors: -

Agreed and corrected (Page 13, Line 21).

RC1: - Page 12, line 7, the should not be capitalized.

Authors: -

Agreed and corrected. (Pg 14, Line 3)

RC1: - Fig. 1, a picture or diagram of the science payload would be good here.

Authors: -

The same has been added to the revised MS.

RC1: - Fig. 6, the white dot is barely visible and even missing, maybe, in some of the panels. Make it more prevalent to see it

Authors: -

Fig. 6 has been revised as suggested.

RC1: - Fig. 7, what is the horizontal bar from the left axis to ZF2? What is the nitrate observed for a comparison point? Also, red and green (also for other figures) is generally not a good combination for colored blind readers.

Authors: -

Observed nitrate levels have been mentioned in the revised manuscript (P 14, L 1; Fig. 7, P 34).

RC1: - Fig. 8, it was mentioned dilution of CO is observed. The figure makes CO look relatively flat.

Indeed, CO is relatively flat for ZF3 but not for ZF2. The major difference between both trajectory ensembles is notable between 15-17 km for ZF2 where air masses were influenced by convection within the AMA. Elevated CO are observed up 120 ppbv near 60-80 hours and decrease to 80-90 ppbv at the time of the flight. We believe that this drop might be the result of dilution with air masses poor in CO.

RESPONSE TO REVIEWER 2

RC2: - *P2L7-9, 'The sampled air masses in winter 2018 were likely affected by smoke from the Pacific Northwest fire event in Canada, which occurred 7 months prior to our campaign, leading to concentration enhancements of $SO_4^{2-}$ and $Ca^{2+}$.':*

The influence of fires is not detailed in the main text. Please add a paragraph where this finding is explained in more detail as well as adding references.

On P8L11, $Ca^{2+}$ is referred to as an indication for mineral dust and $K^+$ for biomass burning. How does this fit to the statement here, since $K^+$ is not measured in the winter 2018 flight (see e.g. Fig. 2)?

Authors: -

The section on Canadian wildfires is discussed in the revised manuscript (Section 5, Pg 9) and a plot of aerosol extinction from SAGE III/ISS is added in the supplementary material to demonstrate that the sampling took place in the bottom part of aged smoke layer.

Although $Ca^{2+}$ is not considered to be a tracer of biomass burning itself, PyroCbs which resulted in stratospheric smoke may also entrain boundary layer aerosols such as dust. To our knowledge, this is the first chemical analysis of aerosols from an aged stratospheric smoke plume and thus results may not be consistent with the expected "tropospheric smoke" composition.

RC2: - *P3L3, 'World Health Organization recommendations':*

Please add a reference.

Authors: -

Agreed and corrected.

RC2: - *P3L19:* You may add here '(Wagner et al., 2020)'

Authors: -

The suggested citation has been added in the revised manuscript (Page 3 Line 9).

RC2: - *P4-5, chapter '1.3 What is known about ATAL's composition?':*

For the sake of completeness, the different views on mineral dust should also be mentioned, which is either predicted to be the major constituent of the ATAL (Fadnavis et al., 2013; Lau et al., 2018; Yuan et al., 2019; Ma et al., 2019; Bossolasco et al., 2020) or of minor importance (Yu et al., 2015; Gu et al., 2016; Yu et al., 2017; Fairlie et al., 2020).

Authors: - We agree with the suggestion and have included the same in the revised manuscript (Page 4 Line 16).

RC2: - *P5L22, 'It translates into a mass concentration of 40 ng/m3 assuming that the aerosols were liquid sulfate droplets':*

1. '…of around 40…'
2. STP should be mentioned always, if it applies.

Authors: - We agree with the suggestion and have mentioned the same wherever appropriate.

RC2: - *P6L28, Fig. 1, Fig. 2:*

1. Provide exact dates of all flights in the text .
2. The flight abbreviations in Fig. 1 and Fig. 3 are mixed up: in Fig. 1 'ZFW' is the winter 2018 flight while in Fig. 3, it is 'ZF-1'. Please present those in a unique way.
3. From the four flights listed in Fig. 1, only 3 are mentioned in the text. Please explain why.

Authors: -

Corrected as suggested. Kindly refer to the revised version of the MS (P 7, Line 2 for flight dates)

Fig. 1 has been modified accordingly (P 28).

 Finally, out of the four ZF flights, flight ZF1 was a test flight launched to give us a clear idea of floating concept using ballast. The same has been mentioned on P 6, L 25.

RC2: - *P8L10-11, 'with traceable amounts of proxies for mineral dust ($Ca^{2+}$) and biomass burning ($K^+$).'*
 Please provide references for this statement and explain it a bit more.

Authors: -

Presence of non-sea-salt-$Ca2+$ in aerosols is often used as a proxy for mineral dust (Schüpbach et al., 2013), and non-sea-salt-$K+$ in aerosols is a proxy for biomass burning (Li et al., JGR, 2003). Although their concentrations were too low (close to the detection limit), their presence indicates a possibility of traces from mineral dust and biomass burning. These lines along with the references are added in the revised manuscript (P 8, L 6).

RC2: - *P8L20, '...but did find the same in the flight samples of winter (Fig. 3).':*

Please formulate this sentence clearer. As it is written, one could think that ammonium has been found during the winter flight, which is not the case according to Fig. 3.

Authors: -

The corresponding statement has been revised and rearranged (Pg 8, Line 13). Please refer to the revised manuscript.

RC2: - *P9L9, 'observations from the CALIOP lidar onboard the CALIPSO satellite.':*

How has the CALIOP data been averaged? Which co-incidence criteria have been applied? Has any cloud-clearing been used.

The CALIOP data were averaged between 17 and 18N along the orbit track which passed within 100 km of the balloon flight trajectory. No cloud-clearing of the data have been applied. Figure S3 added to the manuscript shows that the layer observed between 15-17 km is associated with particulate depolarization and color ratio consistent with a sub-visible cirrus cloud. The text in the manuscript has been changed accordingly (P 9, L16).

RC2: - *P9L14, 'likely made of aspherical particles':*

Does this mean that during this part of the balloon flight, air has been sampled within a cirrus cloud? If so, please state this clearly and any implications this might have on the analysis.

Authors: -

The derived particulate depolarization ratio from CALIOP level 2V4.1 within the layer is 0.47+/-0.06 (Fig. S3) and optical depth 0.03+/0.02 consistent with a subvisible cirrus cloud. Fig. 4 indicates that flight ZF2 sampled within two different air masses. The first being within an ice cloud and the second in a cloud-free region. A discussion on the implications this flight had is discussed in the revised MS (P 9 L 29 onwards).

RC2: - *P9L18, 'was done for more than 1 h in a cloud-free region enhanced with aerosols above.':*

Please formulate this clearer – one could interpret it as if the aerosols have been above the balloon.

Authors: - The statement has been rearranged for clarity (P 9, Line 17).

RC2: - *P9L21-24:*

Is this a general comment on cirrus above Gadanki or does it refer to the situation during one of the flights. Please be clearer here.

Authors : - The section has been revised. Please refer to the corresponding section in the revised version of the manuscript (P 10, Line 11).

RC2: - *9L24-26, 'Moreover, the increasing fraction of sub-visible cirrus clouds between 1998-2003 probably modified on the temperature and the water vapor budget in the Tropical Tropopause Layer (Pandit et al., 2015).'*

There is something wrong with this sentence.

Authors: - The corresponding text has been revised, please refer to the revised manuscript (P 10, Line 11).

RC2: - *P10L1, 'trajectories (black lines) and deep convection influence 1 (red dots).'*

I cannot see this in Figure 5.

Authors: - corrected in the revised manuscript (Pg 10, Line 22).

RC2: - *P11L4, 'CO, nitrate, sulfate, and black carbon (BC) aerosol concentrations':*

1. To support a discussion on mineral dust and to compare with the observed $Ca^{2+}$, it would be good to show here also the model perspective.

2. Please also provide the model maps of $NH4^+$ and discuss differences between model and the lack of ammonium in the observations.

Authors: Dust and NH4+ results from GEOS-Chem have been added in Fig.6.

RC2: - *P11L11, 'nitrate are significantly lower':*

Please provide here numbers (…% lower). From Fig. 7 it seems that these concentrations are not a much lower.

Authors: - Corrected as suggested (P13, Line 12).

RC2: - *P11L20, 'The NOx lifetime is believed to increase downwind from the outflow':*

Please provide a reference for this statement.

Authors: - Please refer to the revised version of the manuscript (Page 13, Line 17).

RC2: - P11L31-P12L4, 'Fig.5 shows that GEOS-chem could simulate convective activities reaching levels between 14-15 km …':

Please describe more clearly where this is the case. Fig. 5 does not reach down to the ground, so one cannot judge if the transported air stems from the boundary layer. Further, there are cases with indications of convection from HIMAWARI which are not captured by the model (ZF2, highest altitude at around 105 deg East).

Authors: - After verifying the vertical evolution of the back-trajectories initialized near 14-15 km for both ZF2 and ZF3 (green color/Fig.5), it appears that the model was able to reproduce the convective transport from the mid-troposphere (9-12 km) to the upper troposphere (14-15 km) (not shown). However, there were indeed no indications for those air parcels to be transported from the boundary layer as initially expected. While it is very likely that convective transport was indeed reproduced in MERRA-2 since it was observed through HIMAWARI-8, the mixture between horizontal and vertical transport could not be visualized through the trajectory calculations. Nevertheless, the only way to explain this rapid ascent of those air parcels is to invoke tropical convection since there are no other transport mechanisms that would explain such a quick ascent. Slow transport by radiative heating would take several days for air parcels to move from the middle to the upper troposphere. We have thus modified the text accordingly (P 16, L 33).

RC2: - P12L11, 'We note that sulfate along the trajectories influenced by Chinese pollution during ZF2 increase significantly…':

1. What means 'significantly' here? Please provide numbers (by …%).
2. Please provide possible explanations (including references) why sulfate is modelled too high compared to the measurements.

Authors: - The suggested changes have been made. Please refer to the revised version of the manuscript (P 14, L 30).

RC2: - *P12-13, chapter '7. Summary and Conclusions':*

1. Please discuss also the representativeness of these balloon observations lying more at the border of the AMA for the ATAL as a whole.
2. Please explain also the relevance of your findings on $Ca^{2+}$ wrt the question of mineral dust as a major constituent of the ATAL (see comment and references above).
3. Please discuss also your finding of high concentrations of nitrite. What could be a relevant mechanism for its production?

Authors: Changed accordingly in the manuscript.

RC2: - *P13L8-10, 'parcels, the model ability to simulate convective influence at higher altitudes seem to be limited.':*

This is not mentioned in the main text – please explain it, where the influence of convection and the ability of the model to simulate convection is discussed.

Authors: - Please refer to the revised manuscript for the suggested explanation (P 17, L 9).

RC2: - *P13L17, 'where smaller nitrate particles were found which could also indicate the influence of new particle formation.':*

Also this is not discussed in the main text. Please provide an explanation where the relevant Figure is explained including possible references.

Authors: Discussed on P 10, L 2).

TECHNICAL COMMENTS: -

**Technical comments:**

*RC2: -P1L39, 'STP':*

Explain abbreviation.

Authors: - Corrected as suggested (P1L39).

RC2: - *P2L5, 'with particle size radius (0.05-2µm)':*

1. 'with particle size radii from 0.05 to 2 µm'
2. Here and all over the text: there should be a space between number and unit

Authors: - corrected as suggested (P2, L5).

RC2: - *P2L6, 'mass':*

-> 'masses'

Authors: - corrected as suggested (P2, L6).

RC2: - *P2L37, 'have':*

-> 'has'

Authors: - corrected as suggested (P3, L4).

*P3L25, 'of ATAL':*

-> 'of the ATAL'

Authors: - corrected (P4, L8)

*RC2 : - P5L6, 'Höpfner et al., 2016':*

-> 'Höpfner et al., 2019'

Authors: - Corrected (P4, L16).

RC2: - *P5L21, 'a concentration of 20 particle/cm3':*

-> '… of about 20 …'

Authors: - Accept & Corrected (P5, L15).

RC2: - *P6L26, ', to':*

-> '. To'

Authors: - Sentence has been reworded (P6, L26).

RC2: - *P8L14, 'Ca$_2^+$':*

-> 'Ca$^{2+}$'

 Authors: - Corrected (P8, L11)

RC2: - *P8L19+22, 'Hopfner':*

-> 'Höpfner'

 Authors: - Corrected (P4, L16+31).

RC2: - *P9L16, ', pressure':*

-> ', the pressure'

 Authors: - Corrected as suggested (P6, L19).

RC2: - *P10L8, 'We conduct GEOS-Chem':*

-> 'We have conducted GEOS-Chem'

 Authors: - Accept & corrected (P 12, L2).

RC2: - *P10L30, '2.5º by 2º horizontal':*

-> 'latitude x longitude' or 'longitude x latitude' ?

Authors: - corrected accordingly (P12, L24).

RC2: - *P11L12, 'is':*

-> 'are'

Authors: - The targeted sentence has been revised. (P 13, L3-5).

RC2: - *P12L11, 'increase':*

-> 'increases'

Authors: - Corrected (P 14, L11).

RC2: - *P12L13, 'NO3':*

-> 'NO$_3^-$'

Authors: - Corrected (P 14, L4).

RC2: - *P12L20, 'onboard':*

-> 'aboard' or 'on board'

Authors: - Corrected accordingly (P 4, L15).

RC2: - *P14-20, refernces:*

doi' s are missing for some the references, please check.

Authors: - We agree with the suggestion and have included the same.

RC2: - *P25, Fig. 4:*

Why are only positive ascent rates given in the color scale?

Authors: - Since providing the ascent rates does not furnish any additional information to the existing figure, we decided to skip the same.

RC2: - *P26, Fig. 5:*

x-axis title missing in bottom right panel.

Authors: - Agreed and corrected.

.

---

## Author Response (AR2)

**Response to second review**

We would like to thank reviewer #1 for those helpful suggestions and comments which will help improving this manuscript.

Vernier et al. evaluated aerosol data collected on balloons launched into the Asian tropopause region during flight intensive flights (and one test flight). The authors generally have done a good job addressing many of the concerns raised by both reviewers, improving the paper. However, there are still some comments, as listed below, the authors should address before the paper should be accepted to ACP.

1) Page 2, Line 15 - 16: Please rephrase to "since NAT has already been observed in the tropical upper tropopsphere and lower stratosphere in other studies."

This is now corrected.

2) Page 7, Line 17: remove the m in prepared

Corrected.

3) Page 8, Line 6 - 7: Please rephrase to "two flights samples collected during the summer 2017 campaign (ZF2, 15th Aug. and ZF3, 21st Aug.), in comparison"

Corrected.

4) Pag 9, Line 21 - 23: Please rephrase to "The derived particulate depolarization ratio from CALIOP level 2v4.1 within the layer was 0.47+/-0.06 (Fig. S3) and was associated with an optical depth of 0.03+/-0.02, indicating the presence of a subvisible cirrus cloud."

Corrected.

5) Section 4.1. I am slightly confused by the reasoning here that $HNO_3$ is transported into the UTLS instead of being locally produced. E.g., Bela et al. (2016) found nearly 90% scavenging efficiency for $HNO_3$ in convection. Reconciliation of this high scavenging efficiency needs to be addressed.

Yes, however scavenging efficiency reported by Bela et al. (2016) was based on aircraft measurements over the continental United-States in the upper troposphere near 10-12 km while our measurements took place at higher altitudes in more polluted conditions where the transport of NOx is expected to be higher with a potential production of $HNO_3$ in the Upper Troposphere and Lower Stratospheric region. We added a line about this. Section 4.1 has been modified accordingly.

6) Page 10, lines 5 - 6: It is unclear what "buffering process results in nitrate naturalization" means.

Line 5-6 has been removed since it was already discussed in line 25-26 p9.

7) Page 10, line 16 - 17: Please correct the capitalization of In after However

Corrected.

8) In all new text and throughout paper, please check the subscript of numbers and letters after chemical formulas (e.g., HNO2, NO2, NO3, NOx, etc.).

Corrected.

9) Addition of Sect. 5 is greatly appreciated; however, there are still some questions concerning it:
9a) As authors stated, K+ is signature for biomass burning. Why is no K+ observed?

We do not have a full explanation about this. Measurements in PyroCbs smoke plume are extremely rare and the chemistry not fully understood. While K+ is an element indicating the signature of biomass burning, we do not really know if it's still the case 6 months after a fire.

9b) NOx is normally emitted at high concentrations with biomass burning. Why is there no nitrite and nitrate?

We would propose the same argument than above.

9c) Also, it's surprising that SO4 is so high in this plume, when it has generally been observed that SO4 is a minor component of biomass burning aerosol. Why do the authors think SO4 is the major anion?

We believe that the BB plume could have been mixed with stratospheric sulfate which is a dominant component of stratospheric aerosol. In addition, PyroCbs plumes have also shown to contain the signature of tropospheric species and thus, we believe that SO2 as well as dust could have been transported in the UTLS.

10) Page 13, Line 1, please correct (1989 to (1989)

Corrected.

11) Page 13, Line 7: It is unclear what the authors mean that HNO2 undergoes rapid reduction. One of the challenges in measuring gas-phase HNO2, beyond wall loss as the authors note, is it's short lifetime due to photolysis, meaning it generally is at very low concentrations away from point sources. Please clarify.

We modified this sentence and added something along this line.

12) Page 13, line 15 - 16: It is unclear why the authors mention nitrification. Are they suggesting that nitrification in the soils produces enough HNO2 to be observed in the UTLS or that nitrification is occurring in the UTLS?

We argue that the presence of NH3 produced by agricultural activities combined with NOx can be potentially transported in the UTLS by convection and form nitrate.

13) Page 13, Line 24 - 29 and Page 14, Line 1 - 27: This should be in the methods section.

We decided to keep this section here since we believe that the GEOS-Chem simulation and its comparison with our measurements are part of the discussion and would thus read better where it is.

14) Page 15, Line 4 - 5: From the figure, it is unclear that BC is increasing. Instead, it appears BC has decreased in the region of the balloon launches. Please clarify.

Yes, it was a mistake which is now corrected.

15) Fig. 7, it would be beneficial to include a vertical bar for the flights

Now included

16) Page 15, Line 6: correct first

Sorry, we could not understand which correction is suggested here.

17) RC1.3 response. ISORROPIA has been evaluated extensively with in-situ nitrate observations, e.g., Guo et al., 2016, Guo et al., 2017, and Ibikunle et al., 2020, to name a few.

Thanks for pointing those references. They have been included in the manuscript.

References:

Bela et al., Wet scavenging of soluble gases in DC3 deep convective storms using WRF-Chem simulations and aircraft observations, JGR, 2016.

Guo et al., Fine particle pH and the partitioning of nitric acid during winter in the northeastern United States, JGR, 2016.

Guo et al., Fine particle pH and gas-particle phase partitioning of inorganic species in Pasadena, California, during the 2010 CalNex campaign, ACP, 2017.

Ibikunle et al., Fine particle pH and sensitivity to NH3 and HNO3 over summertime South Korea during KORUs-AQ, ACPD, 2020.